# Private Evolution Converges

**Tomas Gonzalez**
Carnegie Mellon University
tcgonzal@andrew.cmu.edu

**Giulia Fanti**
Carnegie Mellon University
gfanti@andrew.cmu.edu

**Aaditya Ramdas**
Carnegie Mellon University
aramdas@cs.cmu.edu

## Abstract

Private Evolution (PE) is a promising training-free method for differentially private (DP) synthetic data generation. While it achieves strong performance in some domains (e.g., images and text), its behavior in others (e.g., tabular data) is less consistent. To date, the only theoretical analysis of the convergence of PE depends on unrealistic assumptions about both the algorithm's behavior and the structure of the sensitive dataset. In this work, we develop a new theoretical framework to understand PE's practical behavior and identify sufficient conditions for its convergence. For $d$-dimensional sensitive datasets with $n$ data points from a convex and compact domain, we prove that under the right hyperparameter settings and given access to the Gaussian variation API proposed in [33], PE produces an $(\varepsilon, \delta)$-DP synthetic dataset with expected 1-Wasserstein distance $\tilde{O}(d(n\varepsilon)^{-1/d})$ from the original; this establishes worst-case convergence of the algorithm as $n \to \infty$. Our analysis extends to general Banach spaces as well. We also connect PE to the Private Signed Measure Mechanism, a method for DP synthetic data generation that has thus far not seen much practical adoption. We demonstrate the practical relevance of our theoretical findings in experiments.

## 1 Introduction

Many modern machine learning applications rely on user data, making data privacy protection a central concern. In this regard, differential privacy (DP) [17, 19] has become a *de facto* standard for safeguarding sensitive information of individuals. Given a private dataset, many problems—including regression [40, 12], deep learning [1], and stochastic optimization [6, 2]—can be performed in a DP manner. While it is possible to design a different DP algorithm for each specific task, a reasonable alternative is to generate a DP *synthetic* dataset that preserves many of the statistical properties of the sensitive dataset. By the post-processing property of DP [19], the DP synthetic data can then be fed into existing non-DP algorithms without incurring additional privacy loss; this avoids modifying existing ML pipelines and enables data sharing with third parties (e.g. for reproducibility).

Recently, [33] introduced Private Evolution (PE), a promising new framework for DP synthetic data generation that relies on public, pretrained data generators [48, 32, 27, 41, 50, 28]. PE is currently competitive with—and sometimes improves on—state-of-the-art models in terms of Fréchet inception distance (FID) and downstream task performance in settings such as images and text [33, 48, 32, 27]. In addition, PE is training-free, whereas current state-of-the-art approaches typically train (or finetune) a generative model on the sensitive dataset using DP-SGD [49, 35, 14, 11]. However, in some settings, including tabular data [41], and image data with mismatched distributions [21], PE has achieved limited success. To better understand when PE works, it is crucial to improve our theoretical understanding of the algorithm.

At a high level, PE works as follows (Figure 1). First, it creates a synthetic data set $S_0$ with an API that is independent of the sensitive dataset $S$ (e.g a foundation model trained on public data). Then, iteratively it refines the synthetic data, creating $S_1, S_2, ...$, where $S_t$ is obtained from $S_{t-1}$ by

39th Conference on Neural Information Processing Systems (NeurIPS 2025).

generating variations $V_t$ of the dataset $S_{t-1}$ and then privately selecting the samples that are closest to $S$. Doing so, the synthetic datasets gets 'closer' to $S$ over time.

In [33], the authors study a theoretically tractable variant of PE, analyzing its convergence in terms of the Wasserstein distance, with the aim of understanding its behavior and justifying its empirical success. However, this theoretically tractable version of PE differs in many significant ways from the algorithm that is used in experiments; further, the convergence analysis in [33] makes the unrealistic assumption that every point in $S$ is repeated many times. We start by showing that the proof technique in [33] is inherently limited by this multiplicity assumption; if we remove the assumption, then PE can only be $\varepsilon$-DP with a parameter $\varepsilon$ that scales as $d \log(1/\eta)$, where $d$ is the ambient dimension

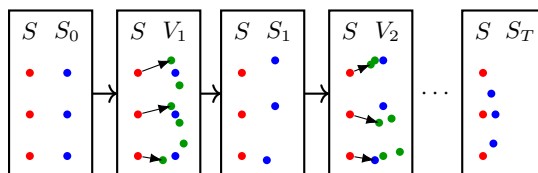

Figure 1: High-level illustration of private evolution (PE). $S$ represents the sensitive data, shown in red. $S_t$ are the synthetic datasets, shown in blue, and are created with the variations from $V_t$ (in green) that are closest to $S$.

of the data and $\eta$ is the final accuracy of the algorithm in terms of the Wasserstein distance. This is impractical in even moderate dimensions.

In this paper, we address the limitations in prior analysis of PE by providing a convergence analysis that does not require strong assumptions about the nature of the dataset and more closely matches the PE algorithm used in practice. We formally prove worst-case[1] convergence guarantees in terms of the 1-Wasserstein distance. We provide an informal version of our Theorem 4.1 below.

**Theorem 1.1** (Convergence of PE (Informal)). *Consider a data domain $\Omega \subset \mathbb{R}^d$ with $\ell_2$ diameter $D$. For any dataset $S \in \Omega^n$ and $0 < \varepsilon, \delta < 1$, there exist APIs and a parameter setting such that PE (Algorithm 2) is $(\varepsilon, \delta)$-DP and it outputs a synthetic dataset $S'$ satisfying*

$$\mathbb{E}[W_1(\mu_S, \mu_{S'})] \leq \tilde{O}\left(dD\left(\frac{\sqrt{\log(1/\delta)}}{n\varepsilon}\right)^{1/d}\right),$$

*where $\mu_S$ is the empirical distribution of the dataset $S$ and similar for $S'$, and $W_1$ is the 1-Wasserstein distance between distributions (Definition 2.2).*

As a corollary, we find sufficient conditions on the APIs used by PE to understand its convergence in more general settings, such as Banach spaces. We also identify strong connections between PE and the Private Signed Measure Mechanism (PSMM), an algorithm for DP synthetic data generation under pure DP [25] that has theoretical guarantees. Hence, our work bridges the theory and practice of PE in both directions: we provide theory for a practical version of PE, and show how PE naturally arises as a practical version of PSMM.

**Contributions** Our work significantly improves the theory of PE presented in [33], providing a more realistic theoretically tractable version of PE and eliminating unrealistic assumptions in the convergence analysis. More concretely, we make the following contributions.

- We prove a lower bound (Lemma 3.1) that establishes that without the multiplicity assumption on the data, the convergence proof of PE provided in [33] only works for undesirable privacy parameters under pure DP. This indicates that a new analysis for PE is needed.

- We propose a new theoretically tractable variant of PE, Algorithm 2. Under this model, we formally prove worst-case convergence rates with respect to the 1-Wasserstein distance as $|S| \to \infty$; see Theorem 4.1. We identify sufficient conditions for convergence of PE in more general settings such as Banach spaces (Appendix G).

- We draw connections between PE and the Private Signed Measure Mechanism (PSMM), an algorithm for DP synthetic data generation [25]. We exploit this connection by using tools from the analysis of PSMM to prove the convergence of PE. Finally, we also show how PE arises naturally when trying to make PSMM 'practical'. See Section 5 for details.

---

[1]Note that convergence is proven in the worst-case over problem instances for a specific set of hyperparameters and APIs.

- Our theory offers an explanation for phenomena observed in prior practical applications of PE—such as its sensitivity to initialization—and offers guidance for future use, including principled parameter selection based on theoretical insights. See Section 6 for details.

## 2   Preliminaries

**Definition 2.1** (Differential Privacy [17]). *A randomized algorithm $\mathcal{A}$ is $(\varepsilon, \delta)$-differentially private if for any pair of datasets $S$ and $S'$ differing in at most one data point and any event $\mathcal{E}$ in the output space, $\mathbb{P}[\mathcal{A}(S) \in \mathcal{E}] \leq e^{\varepsilon}\mathbb{P}[\mathcal{A}(S') \in \mathcal{E}] + \delta$.*

**Definition 2.2** (1-Wasserstein distance [44]). *Let $(\Omega, \rho)$ be a metric space and $\mu, \nu$ two probability measures over it. The $1$-Wasserstein distance between $\mu, \nu$ is defined as*

$$W_1(\mu, \nu) = \inf_{\gamma \in \Gamma(\mu, \nu)} \int_{\Omega \times \Omega} \rho(x, y) d\gamma(x, y),$$

*where $\Gamma(\mu, \nu)$ is the set of distributions over $\Omega^2$ whose marginals are $\mu$ and $\nu$, respectively.*

Next, we formally introduce the problem of DP Synthetic Data studied in this paper.

**Definition 2.3** (DP Synthetic Data). *Let us denote by $\mu_S$ the empirical distribution of a dataset $S$, given by $\frac{1}{|S|}\sum_{z \in S}\delta_z$. Let $(\Omega, \rho)$ be the data metric space. Then, given a dataset $S$ containing sensitive information, the problem of DP Synthetic Data generation consists of designing an $(\varepsilon, \delta)$-DP algorithm $\mathcal{A} : \Omega^n \to \Omega^m$ that returns a DP synthetic dataset $S'$ with the property that $W_1(\mu_S, \mu_{S'})$ is small either in expectation or with high probability, with respect to the randomness in $\mathcal{A}$.*

In the past, there have been different theoretical formulations of the DP Synthetic Data problem (see Section A). In most of them, a fixed set of queries $\mathcal{Q}$ is used to measure the accuracy of the synthetic data: the answer to a query $q \in \mathcal{Q}$ should be similar when querying the sensitive dataset $S$ and the synthetic dataset $S'$. In other words, $\max_{q \in \mathcal{Q}} |q(S) - q(S')|$ should be small. By Kantorovich duality [44], we can alternatively define the 1-Wasserstein distance as

$$W_1(\mu_S, \mu_{S'}) = \sup_{f \in \mathcal{F}_{BL}} |\mathbb{E}_{Z \sim \mu_S}[f(Z)] - \mathbb{E}_{Z \sim \mu_{S'}}[f(Z)]|,$$

where $\mathcal{F}_{BL} = \{f : \Omega \to \mathbb{R} : \|f\|_\infty \leq \mathrm{diam}(\Omega) \text{ and } f(z_1) - f(z_2) \leq \rho(z_1, z_2) \forall z_1, z_2 \in \Omega\}$ is the set of bounded and 1-Lipschitz functions from $\Omega$ to $\mathbb{R}$. Hence, in Definition 2.3 $W_1(\mu_S, \mu_{S'})$ being small implies that the synthetic dataset approximately preserves the answers of $S$ to all Lipschitz queries, making the synthetic dataset $S'$ useful for any downstream task with Lipschitz loss.

**Notation.**   Samples lie in the metric space $(\Omega, \rho)$. The diameter of $\Omega$ is denoted by $\mathrm{diam}(\Omega) = \sup_{z_1, z_2 \in \Omega} \rho(z_1, z_2)$. $\mathcal{P}(\Omega)$ is the space of probability measures supported on $\Omega$. A dataset is a set of elements from $\Omega$. For a dataset $S$, $|S|$ represents its cardinality, $S[i]$ its $i$-th element and $\mu_S$ the empirical distribution $\frac{1}{|S|}\sum_{z \in S}\delta_z$, where $\delta_z$ is the dirac delta mass at $z$. The bounded Lipschitz distance between the signed measures $\mu, \nu$ supported in $\Omega$ is $D_{BL}(\mu, \nu) := \sup_{f \in \mathcal{F}_{BL}} \int f du - \int f dv$. For probability measures $\mu, \nu$, $D_{BL}(\mu, \nu) = W_1(\mu, \nu)$. $\Delta_d$ is the standard probability simplex in $\mathbb{R}^d$. $v[i]$ indicates the $i$-th coordinate of a vector $v \in \mathbb{R}^d$. $\mathtt{nint}(x)$ denotes the nearest integer to $x$.

## 3   Background: Private Evolution (PE) and Prior Theory

Lin *et al.* present the only prior convergence analysis of PE [33]. Their analysis applies to a variant of PE (that we call *theoretical PE*), which is outlined in Algorithm 1. Below, we first describe Algorithm 1 and explain how it differs from how PE is used in practice (Algorithm 3, which we call *practical PE*). Then, we explain the limitations of prior analysis.

PE (both the practical variant, and the theoretical variant in Algorithm 1) generates a DP synthetic dataset by iteratively refining an initial random dataset using a DP nearest-neighbor histogram (Algorithm 4). To do so, it requires access to two APIs that are independent of the sensitive dataset $S$, $\mathrm{Random\_API}$ and $\mathrm{Variation\_API}$[2]. For our theory, we use the same theoretical API models proposed in [33], which we detail below.

---

[2]In practice, generative models trained on public data and simulators can be used as APIs.

---

**Algorithm 1** (Theoretical) Private Evolution; steps in blue differ from practical PE [33]

---

1: **Input:** sensitive dataset: $S \in \Omega^n$, number of iterations: $T$, noise multiplier: $\sigma$, distance function: $\rho(\cdot, \cdot)$, multiplicity parameter $B \geq 1$, threshold $H > 0$
2: **Output:** DP synthetic dataset: $S_T \in \cup_{m \in \mathbb{N}} \Omega^m$
3: $S_0 \leftarrow$ Random_API$(n)$
4: **for** $t = 1 \ldots T$ **do**
5:  $V_t \leftarrow$ Variation_API$(S_{t-1})$
6:  $\hat{\mu}_t \leftarrow$ NN_histogram$(S, V_t, \rho) \in \Delta_{|V_t|}$ (see Alg. 4)
7:  $\tilde{\mu}_t \leftarrow \hat{\mu}_t + \mathcal{N}(0, \sigma^2 I_{|V_t|})$
8:  $\mu'_t[i] \leftarrow \tilde{\mu}_t[i] \mathbb{1}_{(\tilde{\mu}_t[i] \geq H)}$ for every $i \in [|V_t|]$
9:  $S_t \leftarrow \cup_{i \in [|V_t|]} S_i$, where $S_i$ is a multiset containing $V[i]$ $\texttt{nint}(n\mu'_t[i]/B)B$ times
10: **end for**
11: **return** $S_T$

---

- Random_API$(n_s)$ returns $n_s$ data points from the same domain as $S$. In practice, it might generate random samples from a pretrained foundation model. In Algorithm 1, Random_API$(n_s)$ returns $n_s$ data points sampled according to any distribution in $\mathcal{P}(\Omega^{n_s})$.

- For a dataset $S_t$, Variation_API$(S_t) = \cup_{z \in S_t}$ Variation_API$(z)$, where Variation_API$(z)$ returns a set of variations of a data point $z$. In practical PE, a variation of $z$ is another point $z'$ which is close to $z$ in some logical sense. In the case of images, $z'$ could be an image with a similar embedding to that of $z$. However, in Algorithm 1, Variation_API$(z)$ returns a set of variations of $z$ calculated as:

$$\{z\} \cup \left( \cup_{l \in \{1, \ldots, \lceil \log_2(\text{diam}(\Omega)/\alpha) \rceil\}, k \in [2]} \{\text{proj}_\Omega(z + N^{k,l})\} \right), \tag{1}$$

  where $\alpha > 0$ is a small constant and $N^{k,l} \overset{iid}{\sim} \mathcal{N}(0, \sigma_l^2 I_d)$ with $\sigma_l = \frac{\alpha 2^{l-1}}{\sqrt{\pi}[(\sqrt{d} + \log(2))^2 + \log(2)]}$. Note that noise is added at different scales. This is key for the theory to work, because it allows to 'explore' the data domain more effectively. Recall that for a dataset $S_t$, Variation_API$(S_t) = \cup_{z \in S_t}$ Variation_API$(z)$. That is, the API generates $O(|S_t| \log(\text{diam}(\Omega)/\alpha))$ variations by adding independent, spherical Gaussian noises with increasing variances to each sample $z$.

In Algorithm 1, the initial synthetic dataset $S_0$ is created with Random_API. The algorithm then iteratively creates synthetic datasets $S_1, S_2, \ldots$ as follows. At each iteration, variations $V_t$ are created from the current synthetic dataset with Variation_API, and a nearest neighbor histogram indicating how many elements from $S$ have a certain variation from $V_t$ as nearest neighbor. This histogram is privatized by adding Gaussian noise. After thresholding small entries, a new synthetic dataset $S_t$ is constructed *deterministically* by including variations with multiplicity proportional to the noisy histogram (Line 9). This process repeats for $T$ steps, gradually aligning the synthetic data with the private dataset $S$.

Practical PE and Algorithm 1 differ in significant ways, which are highlighted in blue in Algorithm 1. Most notably, Algorithm 1 creates the next synthetic dataset $S_t$ deterministically by adding variations to the next dataset proportionally to the entries of the DP histogram. In practical PE, the next synthetic dataset is instead constructed by sampling with replacement from a distribution defined by the histogram. This seemingly small difference is important for their proof technique, discussed next. More differences between the two variants of PE are discussed in Appendix C.

**Limitations of the utility analysis of Algorithm 1 in [33].** The theoretical analysis in [33] relies on first showing that noiseless PE (Algorithm 1 with $\sigma = 0$) converges, then arguing that when each data point from the sensitive dataset $S$ is repeated $B$ times, with high probability the noisy version behaves like noiseless PE for large $B$.

In the noiseless case the elements of $S$ and $S_0$ are matched: assign $S_0[i]$ to $S[i]$. Then iteratively, for each $i$, $S_t[i]$ is chosen as the element from Variation_API$(S_{t-1})$ that is closest to $S[i]$, which is likely to be closer to $S[i]$ than $S_{t-1}[i]$. After enough steps, $\rho(S[i], S_T[i]) \leq \eta$ for all $i \in [n]$; we say the datasets are $\eta$-close, written as $S =_\eta S_T$. To handle noise, [33] assumes every real data point is repeated $B$ times. This boosts the signal in the noisy step (Line 7), allowing the algorithm to behave

like the noiseless version. In other words, although the proof assume each data point is repeated $B$ times, the noise scale provides a DP guarantee for neighboring datasets that differ in a *single* sample. This setup is unrealistic and sidesteps the core challenge of differentially private learning. More details in Appendix C.

We provide a formal lower bound giving evidence that the proof technique in [33] is fundamentally limited, in the sense that their unrealistic assumption is necessary for their proof technique to work. More concretely, in Lemma 3.1 we show that if we remove the multiplicity assumption, then any $\varepsilon$-DP algorithm can only output a DP synthetic dataset $S_T$ that is $\eta$-close to $S$ when $\varepsilon = \Omega(d \log(1/\eta))$, or equivalently, $\eta = \Omega(e^{-\varepsilon/d})$. Proof in Appendix H.

**Lemma 3.1** (Lower bound for $\eta$-closeness). *Consider a metric (data) space $(\Omega, \rho)$ and a fixed dataset $S \in \Omega^n$. Let $\eta > 0$ and $\mathcal{A} : \Omega^n \to \Omega^n$ be such that $\mathbb{P}_{\mathcal{A}}\big[S =_\eta \mathcal{A}(S)\big] \geq 1 - \tau$ for some $\tau \in (0, 1/4)$. Suppose $\mathcal{A}$ is $(\varepsilon, \delta)$-DP for some $\delta \in (0, 1 - 3\tau)$. Then, denoting $\mathcal{M}(\Omega, \rho; 2\eta)$ the $2\eta$ packing number of $\Omega$ w.r.t $\rho$, we must have*

$$\varepsilon \geq \log(\mathcal{M}(\Omega, \rho; 2\eta)).$$

**Remark 3.1.** *The packing number $\mathcal{M}(\Omega, \rho; 2\eta)$ is of the order $(1/\eta)^d$ when $\Omega \subset \mathbb{R}^d$ and $\mathrm{diam}(\Omega) \leq 2$ (see Lemma $5.5$ and Example $5.8$ in [45]).*

# 4    Convergence of Private Evolution

We have established several limitations of prior utility analysis of PE: the algorithmic variant of PE analyzed in [33] differs from what is done in practice, and the analysis itself is unlikely to extend beyond very weak privacy regimes, as evidenced by Lemma 3.1. In this section, we introduce a new theoretically tractable version of PE along with its utility convergence guarantees. The algorithm we analyze more closely reflects how PE works in practice and is amenable to worst-case utility analysis.

## 4.1    Convergence of PE in Euclidean space

We start by presenting the details of the algorithm that we analyze. Consider a metric (data) space $(\Omega, \rho)$ and a private dataset $S \in \Omega^n$. Even though we will use the notation $(\Omega, \rho)$ throughout this section, we assume in this subsection that $\Omega \subset \mathbb{R}^d$ is a convex and compact with $\mathrm{diam}(\Omega) \leq D$ and $\rho(\cdot, \cdot)$ is the $\ell_2$ distance. We also assume Algorithm 2 uses the same APIs as Algorithm 1, as described in Section 3.

---

**Algorithm 2** Private Evolution with $D_{BL}$ projection; steps in blue differ from practical PE [33]

---

1: **Input:** sensitive dataset: $S \in \Omega^n$, number of iterations: $T$, number of generated samples: $n_s$, noise multiplier: $\sigma$, distance function: $\rho(\cdot, \cdot)$
2: **Output:** DP synthetic dataset: $S_T \in \Omega^{n_s}$
3: $S_0 \leftarrow \mathrm{Random\_API}(n_s)$
4: **for** $t = 1 \ldots T$ **do**
5:     $V_t \leftarrow \mathrm{Variation\_API}(S_{t-1})$
6:     $\hat{\mu}_t \leftarrow \mathrm{NN\_histogram}(S, V_t, \rho) \in \Delta_{|V_t|}$ (see Alg. 4)
7:     $\tilde{\mu}_t \leftarrow \hat{\mu}_t + \mathcal{N}(0, \sigma^2 I_{|V_t|})$
8:     $\mu'_t \in \arg\min_{\mu \in \Delta_{|V_t|}} D_{BL}(\tilde{\mu}_t, \mu)$
9:     $S_t \leftarrow n_s$ samples with replacement from $\mu'_t$
10: **end for**
11: **return** $S_T$

---

Algorithm 2 iteratively refines an initial random dataset $S_0$ using DP nearest-neighbor information. At each iteration, it generates variations $V_t$ from the current synthetic dataset $S_{t-1}$ and computes a histogram $\hat{\mu}_t$ indicating how often each candidate is the nearest neighbor of points in the sensitive dataset $S$. Gaussian noise is added to this histogram to ensure differential privacy, resulting in a noisy vector $\tilde{\mu}_t$. Rather than using thresholding like Algorithm 1 in Line 8, this algorithm projects $\tilde{\mu}_t$ onto the probability simplex using the $D_{BL}$ distance in Line 8, yielding a valid distribution $\mu'_t$ over $V_t$ that remains close to the noisy estimate under the BL metric. The next synthetic dataset $S_t$ is created

*randomly* by sampling $n_s$ points from $V_t$ according to $\mu'_t$ in Line 9, as opposed to Algorithm 1 which creates $S_t$ deterministically in Line 9. This process is repeated for $T$ iterations, after which the final synthetic dataset $S_T$ is returned. Note that unlike Algorithm 1, our version receives the number of synthetic data points as input. We provide a comparison between Algorithms 1 and 2 in Appendix C.

**Comparison between Algorithm 2 and practical PE.** Algorithm 2 is nearly identical to the practical PE implementation in [33], differing only in how the noisy histogram $\tilde{\mu}_t$ is post-processed (highlighted in blue). We project $\tilde{\mu}_t$—a signed measure from Gaussian noise added to a histogram—onto $\mathcal{P}(V_t)$ by minimizing the bounded Lipschitz distance, which can be done via linear programming (e.g., Algorithm 2 from [25]). This is crucial for obtaining worst-case convergence guarantees. In contrast, [33] applies a simpler heuristic: thresholding $\tilde{\mu}_t$ at some $H > 0$ and re-normalizing, which is more efficient but can discard all votes if values fall below the threshold, making it unsuitable for worst-case analysis. In Section 4.2, we provide data-dependent accuracy bounds for a related noisy histogram obtained from adding Laplace noise and thresholding. These bounds are vacuous in the worst case but can be much tighter in favorable regimes, e.g., if the private data is highly clustered.

**Main Result: Convergence of PE.** We are now ready to present our main result: an upper bound on expected utility of Algorithm 2. We provide a proof sketch, and the full proof is in Appendix E.

**Theorem 4.1** (Convergence of PE). *Let $(\Omega, \|\cdot\|_2)$ with $\Omega \subset \mathbb{R}^d$ closed and convex and* $\mathrm{diam}(\Omega) \leq D$ *be the sample space. Fix $\sigma \in (0,1)$ and let $\gamma = \frac{1}{8\pi[((\sqrt{d}+\log(2))^2 + \log(2)]}$.* *Suppose $\alpha = D\sigma^{1/\max\{d,2\}}$ in (1), and let $S_T$ be the output of Algorithm 2 run on input* $S \in \Omega^n, T \geq \left\lceil \log\left(\frac{\gamma}{D\sigma^{1/\max\{d,2\}}}\right)/\gamma \right\rceil, n_s = \sigma^{-1}(2\lceil \log_2(D/\alpha)\rceil + 1)^{1/\max\{d,2\}-1}, \sigma, \|\cdot\|_2.$ *Then*

$$\mathbb{E}\left[W_1(\mu_S, \mu_{S_T})\right] \leq \tilde{O}(dD\sigma^{1/d}).$$

*Furthermore, if $\varepsilon, \delta \in (0,1)$, and we instead run Algorithm 2 with the same parameter setting as above except $T = \left\lceil \log\left(\frac{\gamma(n\varepsilon)^{1/\max\{d,2\}}}{\left(4\sqrt{\log(1/\delta)}\right)^{1/\max\{d,2\}}}\right)/\gamma \right\rceil$ and $\sigma = \frac{4\sqrt{T\log(1.25/\delta)}}{n\varepsilon}$, then the algorithm is $(\varepsilon, \delta)$-DP and its output $S_T$ satisfies*

$$\mathbb{E}\left[W_1(\mu_S, \mu_{S_T})\right] \leq \tilde{O}\left(dD\left(\frac{\sqrt{\log(n\varepsilon/\sqrt{\log(1/\delta)})}\log(1/\delta)}{n\varepsilon}\right)^{1/d}\right).$$

Some comments are in order. First, note that $n_s \geq 1$ for $\sigma \in (0,1)$ and to ensure $\sigma \in (0,1)$ under DP we only need $\sqrt{T\log(1/\delta)} \lesssim n\varepsilon$. Second, our analysis suggests that we should set number of evolution steps $T = \tilde{O}(\log(n\varepsilon))$ and the number of synthetic samples $n_s = \tilde{O}(n\varepsilon/\sqrt{T})$, which we use in Section 6.[3] Finally, our rates are vacuous for $d \gtrsim \log(n)$. This is expected when using $W_1$ as a utility metric, as evidenced by the lower bound of $n^{-1/d}$ under pure DP from [9, Theorem 8.5].

*Proof Sketch of 4.1.* Given dataset $S_t$ and variations $V_t$, in iteration $t$, PE constructs: the NN histogram $\hat{\mu}_{t+1}$, the noisy signed measure $\tilde{\mu}_t$ and the projected probability measure $\mu'_t$. Finally, $S_{t+1}$ is sampled from $\mu'_t$. It is possible to prove the following inequality that involves these measures:

$$W_1(\mu_S, \mu_{S_{t+1}}) \leq W_1(\mu_S, \hat{\mu}_{t+1}) + 2D_{BL}(\hat{\mu}_{t+1}, \tilde{\mu}_{t+1}) + W_1(\mu'_{t+1}, \mu_{S_{t+1}}).$$

At a high level, the convergence of PE follows from two facts:

- First, by creating variations of $S_t$ and selecting the closest ones to $S$, the Wasserstein distance to $S$ is reduced. That is, $W_1(\mu_S, \hat{\mu}_{t+1})$ is noticeably smaller than $W_1(\mu_S, \mu_{S_t})$.

- Second, the progress made by creating variations is not affected by the noise from DP and sampling since $D_{BL}(\hat{\mu}_{t+1}, \tilde{\mu}_{t+1})$ and $W_1(\mu'_{t+1}, \mu_{S_{t+1}})$ are both small.

---

[3]In the nonprivate setting ($\epsilon = \infty$), one can skip the noise addition and sampling steps and set $n_s = n$, since $n_s$ does not need to be chosen to balance the error terms that arise from DP. This results in a nonprivate algorithm similar to that of [33] with an expected $W_1$ error of $2\alpha$ after $T \gtrsim d\log(D/\alpha)$ evolution steps without DP constraints, similar to Theorem 1 of [33].

These statements are made rigorous in our proof. Regarding the first statement, Lemmata E.1 and E.2 give, $\mathbb{E}_t[W_1(\mu_S, \hat{\mu}_{t+1})] \leq (1-\gamma)W_1(\mu_S, \mu_{S_t}) + \alpha$ for some $\gamma = \Theta(1/d)$, where $\mathbb{E}_t$ denotes the expectation conditioned on the randomness of the algorithm up to iteration $t$.

To bound $D_{BL}(\hat{\mu}_{t+1}, \tilde{\mu}_{t+1})$, we use the fact that $\tilde{\mu}_{t+1} = \hat{\mu}_{t+1} + Z$ with $Z \sim \mathcal{N}(0, \sigma^2 I_{|V_t|})$, which by definition implies $D_{BL}(\hat{\mu}_{t+1}, \tilde{\mu}_{t+1}) = \sup_{f \in \mathcal{F}_{BL}} \int_\Omega f(d\hat{\mu}_{t+1} - d(\hat{\mu}_{t+1} + Z)) = \sup_{f \in \mathcal{F}_{BL}} \sum_{i \in [|V_{t+1}|]} f(V_{t+1}[i])Z_i$, which is the supremum of a Gaussian process. The expectation of this term can be bounded using empirical process theory; Lemma E.3 states that $\mathbb{E}_t[D_{BL}(\hat{\mu}_{t+1}, \tilde{\mu}_{t+1})] \leq |V_t|\sigma G_{|V_t|}(\mathcal{F}_{BL})$, where $G_{|V_t|}(\mathcal{F}_{BL})$ is a Gaussian complexity term [5]. This term can be bounded using Corollary E.1, which is a consequence of Lemma E.4. We use Dudley's chaining to prove these results. We remark that our technique prove this bound resembles [25], with the difference that they deal with a Laplacian complexity arising from the analysis of a pure-DP algorithm that uses the Laplace mechanism [19].

Finally, to control $W_1(\mu'_{t+1}, \mu_{S_{t+1}})$ we use results from the literature of the convergence of empirical measures in the Wasserstein distance [31, 20] that quantify the Wasserstein distance between a measure and the empirical measure of iid samples from it; see Lemma E.5. We obtain $\mathbb{E}_t[W_1(\mu'_{t+1}, \mu_{S_{t+1}})] \leq \tilde{O}(Dn_s^{-1/\max\{2,d\}})$. Putting all the inequalities together, we conclude that

$$\mathbb{E}_t[W_1(\mu_S, \mu_{S_{t+1}})] \leq (1-\gamma)W_1(\mu_S, \mu_{S_t}) + \alpha + 2|V_t|\sigma G_{|V_t|}(\mathcal{F}_{BL}) + \tilde{O}(Dn_s^{-1/\max\{2,d\}}).$$

Recursing this inequality, we can get a bound on $\mathbb{E}[W_1(\mu_S, \mu_{S_T})]$ while carefully balancing the number of variations with the number of synthetic samples so that the error terms are as small as possible. This finishes the proof of $\mathbb{E}[W_1(\mu_S, \mu_{S_T})] \leq \tilde{O}(dD\sigma^{1/d})$.

Regarding the second result, for privacy we use the fact that $\sigma = \frac{\sqrt{4T\log(1.25/\delta)}}{n\varepsilon}$ is enough to ensure $(\varepsilon, \delta)$-DP of an adaptive composition of $T$ Gaussian mechanisms where each of them adds noise to a NN histogram with $\ell_2$ sensitivity $\sqrt{2}/n$ ([15], Corollary 3.3). For convergence, we plug in the expression for $T$ in $\sigma = \frac{\sqrt{4T\log(1.25/\delta)}}{n\varepsilon}$ and then $\sigma$ in $\mathbb{E}[W_1(\mu_S, \mu_{S_T})] \leq \tilde{O}(d\sigma^{1/d})$. $\qquad\square$

**Remark 4.1.** *Most of our proof techniques work in metric spaces. Under some conditions on the APIs, the analysis of Algorithm 2 can be extended to more general Banach spaces (Appendix G).*

### 4.2 Beyond worst-case analysis

Recall that our Algorithm 2 differs from practical PE in the BL projection onto the space of probability distributions (Line 8). To connect our analysis more to the original PE algorithm, we can swap this BL projection for a different noisy histogram step, which works by thresholding and re-normalizing—similarly to practical PE. We derive a data-dependent bound on the 1-Wasserstein distance between this modified noisy histogram and the non-private one. The bound indicates that in favorable cases, such as when the data is highly clustered, this histogram is closer to the non-private one than the one we would obtain with Gaussian noise and $D_{BL}$ projection, but is looser in the worst case.

In our updated histogram step, we add Laplace noise with scale $2/[n\varepsilon]$ to the entries of the (non-DP) NN histogram that are positive, and then threshold the noisy entries at $H = 2\log(1/\delta)/(n\varepsilon) + 1/n$. More concretely, let $\hat{\mu} = \text{NN\_histogram}(S, V, \rho)$ be the non-DP histogram between a dataset $S$ and a set of variations $V \in \Omega^m$ (see Algorithm 4). The noisy histogram $\tilde{\mu}$ is then given by $\tilde{\mu}[i] = (\hat{\mu}[i] + L_i)\mathbb{1}_{(\hat{\mu}[i]>0, \hat{\mu}[i]+L_i \geq H)}$ where $\{L_i\}_{i \in [m]} \overset{iid}{\sim} \text{Lap}(2/n\varepsilon)$. While this algorithm was proven to be $(\varepsilon, \delta)$-DP in Theorem 3.5 of [43], to the best of our knowledge, its utility guarantee in terms of $W_1$ distance is new. Its proof can be found in Appendix H.

**Proposition 4.1.** *Let $(\Omega, \rho)$ be a metric (data) space and suppose $\text{diam}(\Omega) \leq D$. Given $S \in \Omega^n$ and $V \in \Omega^m$, let $\tilde{\mu}$ be the noisy histogram as described above and $\mu' = \tilde{\mu}/\|\tilde{\mu}\|_1$ if $\|\tilde{\mu}\|_1 > 0$ and $\mu' = \tilde{\mu}$ otherwise. Then, the procedure generating $\mu'$ is $(\varepsilon, \delta)$-DP w.r.t $S$ and for any $\beta \in (0,1)$*

$$\mathbb{E}[W_1(\hat{\mu}, \mu')] \leq \frac{2|\tilde{I}|}{n\varepsilon}L_{|\tilde{I}|}(\mathcal{F}_{BL}) + 2DH(|\hat{I}| + |\tilde{I}|\beta) + \frac{2D\sqrt{2|\tilde{I}|}}{n\varepsilon},$$

where $L_{|\tilde{I}|}(\mathcal{F}_{BL})$ is the Laplacian complexity of $\mathcal{F}_{BL}$, $\tilde{I} = \{i \in [m] : \hat{\mu}[i] > 0\}$ is the set of positive entries in the NN histogram $\hat{\mu}$, $\hat{I} = \left\{i \in \tilde{I} : \hat{\mu}[i] = O\left(H + \frac{\log(|\tilde{I}|/\beta)}{n\varepsilon}\right)\right\}$ is the set of entries in the NN histogram below $O\left(H + \frac{\log(|\tilde{I}|/\beta)}{n\varepsilon}\right)$ and $H = 2\log(1/\delta)/(n\varepsilon) + 1/n$ is the threshold.

**Examples.** To demonstrate when this proposition yields a tighter result, consider the case when $|\tilde{I}|$ and $|\hat{I}|$ are small. As an extreme example, suppose $S$ consists of one data point repeated $n$ times. Then, $|\tilde{I}| = 1$ and $|\hat{I}| = 0$, leading to an error of $O(\frac{D\log(1/\delta)}{n\varepsilon} + \frac{D}{n})$. A similar intuition is valid for highly clustered datasets. On the other extreme, if every positive entry of $\hat{\mu}$ is $1/n$, then $|\tilde{I}| = |\hat{I}| = n$, leading to a trivial error bound. See Appendix F for simulations.

## 5 Connections between PE and Private Signed Measure Mechanism

In this section, we show connections between PE and the Private Signed Measure Mechanism (PSMM), a DP synthetic data method with formal $W_1$ guarantees from [25]. Despite its theoretical guarantees, PSMM is not currently used for high-dimensional, real datasets.

**PSMM as a one-step version of PE.** PSMM takes a dataset $S$ and a partition $\{\Omega_i\}_{i\in[m]}$ of $\Omega$. It privately counts how many points fall into each bin $\Omega_i$, and then builds a DP distribution over $\Omega$ proportional to these counts (details in Appendix I). Synthetic data is sampled from this distribution.

We can reinterpret PSMM in the language of our framework. Instead of a partition, we can provide a discrete support $\Omega' \subset \Omega^m$ such that the Voronoi partition induced by $\Omega'$ coincides with $\{\Omega_i\}_{i\in[m]}$. Counting how many elements fall into each $\Omega_i$ is then equivalent to counting how many points in $S$ have $\Omega'[i]$ as their nearest neighbor. The resulting private distribution can thus be viewed as a DP nearest-neighbor histogram over $\Omega'$. We believe that the reason behind this (modified PSMM) algorithm working is that the NN histogram solves a 1-Wasserstein minimization problem (Proposition 5.1; proof in Appendix H), a property that also plays a key role in our analysis of PE.

Viewed this way, PSMM resembles a single PE iteration where $\Omega'$ plays the role of the variations. If $\Omega'$ is roughly a $(n\varepsilon)^{-1/d}$-net of $\Omega$, PSMM achieves $\mathbb{E}[W_1(S, S')] \le O((\sqrt{\log(1/\delta)}/n\varepsilon)^{1/d})$, improving over PE's bound in Theorem 4.1 by a factor of $d$. However, under the PE setup, where $\Omega'$ must be sampled from $\mathrm{Random\_API}$, this net condition is hard to ensure, explaining PE's slower worst-case convergence.

**Proposition 5.1.** *Consider* $S \in \Omega^n, V \in \Omega^m$. *Let* $\mu^* = \mathrm{NN\_histogram}(S, V, \rho)$ *(see Alg. 4). Then,* $\mu^* \in \arg\min_{\mu \in \Delta_m} W_1\left(\mu_S, \sum_{i\in[m]} \mu[i]\delta_{V[i]}\right)$.

**PE as a practical, sequential version of PSMM.** PSMM's main practical limitation is constructing a discrete support $\Omega'$ that adequately covers $\Omega$. For example, if $\Omega$ represents the space of images, one option is to discretize the pixel space to build $\Omega'$, but this yields unrealistic images. To generate realistic elements, a natural alternative is to sample from a generative model and add images to $\Omega'$ only if they are sufficiently distinct from those already included, aiming to cover as much of $\Omega$ as possible. However, this process can be slow—especially if the generative model places low probability mass to some regions of $\Omega$. Moreover, full coverage of $\Omega$ is unnecessary; we only need to cover the region containing $S$. This motivates a sequential approach: start with an initial estimate $V_0$ of $\Omega'$, and iteratively refine it to better match the support of $S$. This is precisely what PE does—at each step $t$, $V_t = \mathrm{Variation\_API}(S_{t-1})$ can be seen as a refined support estimate, and $S_t$ as a DP synthetic dataset supported on $V_t$ whose empirical distribution approximately minimizes the 1-Wasserstein distance to that of $S$.

## 6 Empirical Results

### Simulations

We begin with simulations where the sample space $\Omega$ is the unit $\ell_2$ ball in $\mathbb{R}^2$ and the sensitive dataset is random over $\Omega \cap \mathbb{R}^2_+$. We use privacy parameters $\varepsilon = 1, \delta = 10^{-4}$. Our theory indicates that

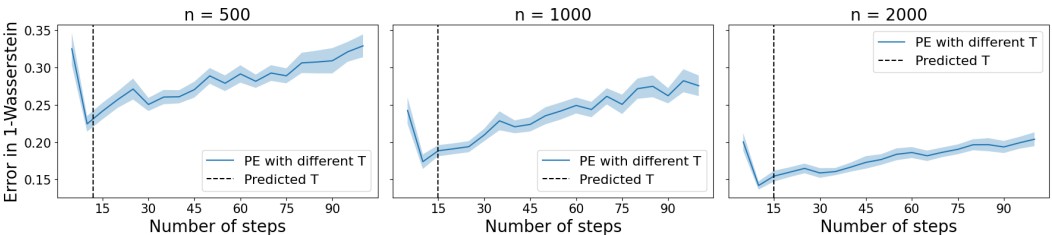

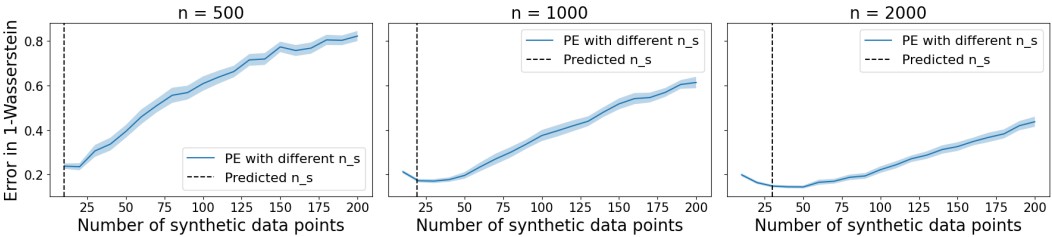

Figure 2: Impact of parameters on PE's performance. Top: performance of the last iterate of PE when run for different number of steps; 'Predicted $T$' marks the theoretically suggested value $T = 2\log(n\varepsilon)$. Bottom: same setup, but replacing $T$ by the number of synthetic samples $n_s$; 'Predicted $n_s$' marks the value of $n_s$ given in Theorem 4.1. We repeat this for different sensitive sample sizes $n$, averaging over 100 runs. The plots illustrate the accuracy of our theoretical predictions.

$O(\log(n\varepsilon))$ PE steps ensure convergence; for simulations we set $T = 2\log(n\varepsilon)$. The noise $\sigma$ is then computed with the analytic Gaussian mechanism [4, Theorem 8]. The remaining parameters are set as in Theorem 4.1. We post-process noisy histograms by truncating at $H = 0$ and then re-normalizing.

**Dependence on initialization.** Let $\Gamma_t \triangleq \mathbb{E}[W_1(\mu_S, \mu_{S_t})]$, i.e. the expected Wasserstein-1 distance between the private data and the $t$th iteration of synthetic data. From equation (2) in the proof of Theorem 4.1, we obtain $\Gamma_T \leq (1 - \gamma)^T(\Gamma_0 - err) + err$ where $err$ increases polynomially with $T$. We note from this equation that if $\Gamma_0 \leq err$, then the optimal number of steps for PE is $T = 0$. However, when $\Gamma_0$ is large, our worst-case upper bound will be more accurate. Figure 5 illustrates this phenomenon. Our findings help to explain prior empirical work. [33] found that PE requires more steps to converge and consistently improves over time when $\Gamma_0$ is large. In contrast, [41] found that in many instances, PE worsens with time, and the optimal number of steps is 1.

**Hyperparameter selection.** Theorem 4.1 suggests setting the number of PE steps $T = O(\log(n\varepsilon/\sqrt{\log(1/\delta)}))$ (independent of $d$). Similarly, the number of synthetic samples $n_s$ should be chosen to balance the error due to adding noise to the NN histogram in Step 7 of Algorithm 2 (which increases with number of synthetic samples) with the error due to sampling with replacement in Step 9 (which decreases with the number of synthetic samples). Figure 2 shows how a suboptimal number of PE steps or synthetic data points degrades data quality.

**Experiments on CIFAR-10**

Next, we evaluate our theoretical predictions on an image modeling task, using two subclasses (Dog and Plane) of images from the CIFAR-10 dataset [30]. We closely follow the experimental setup in [33], using the ImageNet inception embedding [42] and Improved Diffusion [36] trained on ImageNet as $\mathrm{Random\_API}$ and $\mathrm{Variation\_API}$[4]. Unlike [33], we create variations by running the diffusion model with different degrees of variation in each iteration, while [33] fixed the number of variation degrees per iteration and decreased it across iterations. Figure 3 shows that if we run PE with the

---

[4]We use the checkpoint imagenet64_cond_270M_250K.pt from https://github.com/openai/improved-diffusion

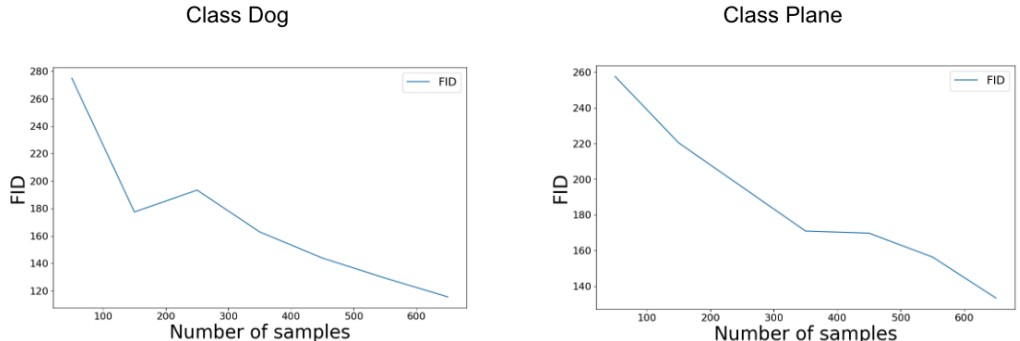

Figure 3: We set the privacy parameters to $\varepsilon = 5, \delta = 10^{-4}$. For each $n \in \{100, 200, ..., 600\}$, we set the hyperparameters according to Theorem 4.1 and run PE. We repeat the experiment 3 times and report the average final FID achieved for each value of $n$, and note that it decreases with larger $n$.

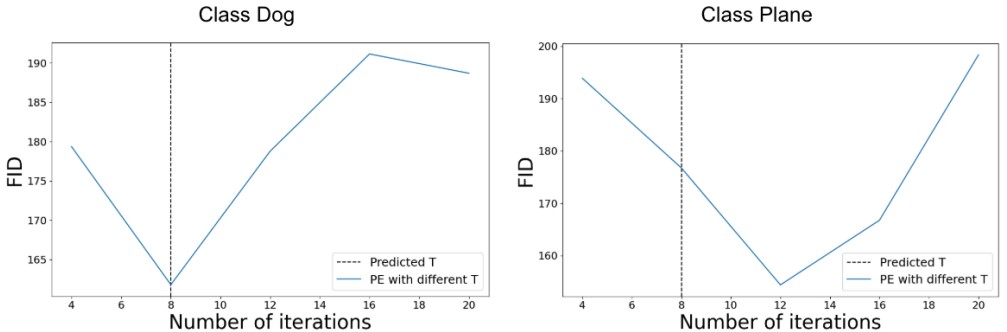

Figure 4: We set the privacy parameters to $\varepsilon = 5, \delta = 10^{-4}$. We consider a fixed number of samples $n = 300$. Then, we set the all hyperparameters according to Theorem 4.1, except the number of steps $T$. For each $T \in \{4, 8, 12, 16, 20\}$, we run PE 3 times and report the average final FID achieved.

hyperparameter setting (including $T$ and $n_s$) derived from Theorem 4.1, then the FID improves (decreases) with more samples. This empirically supports the main claim of our paper: PE converges with enough samples, as long as hyperparameters are set correctly and with appropriate APIs. We also evaluate performance of PE with different choices of $T$, similar to Figure 2. Figure 4 confirms that using the theoretically-informed number of evolution steps $T$ is effective even on real data. In particular, the experiments on CIFAR10 data presented in [33] use $T = 20$, which turns out to be suboptimal, while $T = \log(n\varepsilon)$ is a better choice.

## 7 Conclusion

We identified key algorithmic and analytical issues in prior theory for Private Evolution (PE) and introduced a new theoretical model with formal convergence guarantees. Our version preserves the core logic of practical PE and offers a plausible explanation to behaviors observed by practitioners. The analysis combines empirical process theory, Wasserstein convergence, and properties of APIs. We also connect PE to the Private Signed Measure Mechanism, offering new insights into its theoretical foundations. Overall, our work significantly deepens the formal understanding of PE.

**Limitations and future work**. The choice of Variation_API that we presented in Section 3 is not the same as what practical variants of PE utilize. Furthermore, the convergence rate we proved in Theorem 4.1 suffers from the curse of dimensionality, requiring many samples to ensure convergence. Future work could expand the set of theoretically tractable Variation_API for which we understand the convergence of PE or study alternative accuracy metrics that do not suffer from the curse of dimensionality.

## Acknowledgments and Disclosure of Funding

This work was supported in part by NSF grants CCF-2338772, RINGS-2148359 and CNS-2148359 and the Sloan Foundation. TG was partially funded to attend this conference by the CMU GSA/Provost Conference Funding.

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

# A    Related work

*Private evolution.* There have been multiple follow up works since Lin *et al.* [33] introduced the framework of Private Evolution. [33, 48] propose PE for images and text using foundation models as APIs. Other works have extended the framework in various ways, including adapting it to the federated setting [27], the tabular data setting [41], allowing access to many APIs [50], and replacing the API with a simulator [32]. In a similar vein, [34] proposed zeroth-order optimization methods based on genetic algorithms before [33] in order to handle non-differentiable accuracy measures. Despite many empirical works on PE, to the best of our knowledge, its theoretical properties are studied only in [33], under a number of unrealistic assumptions, as discussed in Section 3.

*Theory of DP synthetic data.* The utility of synthetic data is typically analyzed by quantifying how well the data can be used to answer queries. More concretely, let $\mathcal{Q}$ be a set of queries and $q(S)$ the answer of $q \in \mathcal{Q}$ on sensitive dataset $S$. Most of the theory on synthetic DP data measures the quality of a synthetic dataset $S'$ with $\max_{q \in \mathcal{Q}} |q(S) - q(S')|$[24, 7, 18]. When $\mathcal{Q}$ contains all the bounded Lipschitz queries, then the accuracy measure coincides with $W_1(\mu_S, \mu_{S'})$ (see 2.2); this case was studied in [26, 25, 8]. The cases of sparse Lipschitz queries and smooth queries are studied in [16] and [47], respectively. [22] assumes that $S$ is sampled according to an unknown distribution $\mathcal{D}$ and the accuracy measure is given by $\max_{q \in \mathcal{Q}} |\mathbb{E}_{z \sim \mathcal{D}}[q(z)] - q(S')|$. In this work, we focus on the setting where the accuracy measure corresponds to the Wasserstein-1 distance, i.e., $\mathcal{Q}$ contains all bounded Lipschitz queries. [3] gives a practical algorithm along with accuracy guarantees for privately answering many queries, but only works for datasets from a finite data space.

*Practice of DP synthetic data.* We provide a non-exhaustive selection of practical works on DP synthetic data. For a more comprehensive survey, see [21, 13]. Most practical methods consist of training (either from scratch or finetuning) a non-DP generative model with DP-SGD (that is, the gradients are privatized during training). For example, the following methods have been proposed: DP-GAN [49], G-PATE [35], DP Normalizing Flows [46] and DP Diffusion models [14]. [11] uses the Sinkhorn divergence as a loss function and also uses DP-SGD for training. Several works take inspiration from Optimal Transport (in particular, Wasserstein distances). [37] introduces the Smoothed Sliced Wasserstein Distance, making the loss private (instead of the gradients), and the gradient flow associated with this loss was recently studied in [39]. [38] studied how to generate DP synthetic data under Local DP with entropic optimal transport, and is one of the few practical works which comes with theoretical convergence guarantees.

In general, most practically-competitive synthetic data algorithms do not come with accuracy guarantees, particularly in continuous data space settings, and algorithms with accuracy guarantees tend to be less practical in high dimensions. The bridge between the theory and the practice of DP Synthetic Data transcends PE.

# B    More simulations

First, we provide the figure that shows how our theory captures the dependence on initialization, as explained in Section 6.

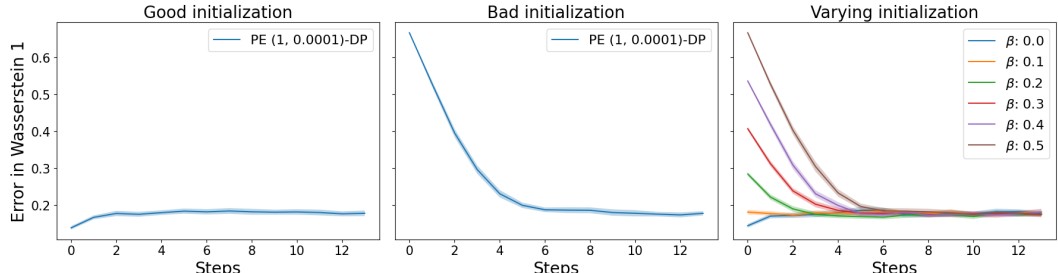

Figure 5: Recall from Section 6 that $\varepsilon = 1, \delta = 10^{-4}$. We set the number of private samples to $n = 1000$. We run PE on different initial datasets created with $\mathrm{Random\_API}$ and plot the accuracy trajectories over $2\log(n\varepsilon) = 12$ iterations. We note that the best number of steps depends heavily on the initialization from $\mathrm{Random\_API}$. The private dataset $S$ is a random dataset in $\Omega \cap \mathbb{R}_+^2$ . Left: When $S_0 = S$ (i.e., $\Gamma_0 = 0$, the private data is the same as the initial synthetic data), the optimal number of PE steps is 0. Middle: When PE is initialized poorly (e.g., $S_0$ consists of only $(0,0)$, so $\Gamma_0$ is large), more iterations are needed. Right: Interpolating between the previous cases parametrized by $\beta$: $S_0 = (1 - 2\beta)S$, PE can improve or degrade performance depending on $\mathrm{Random\_API}$; it never exceeds the worst-case error bound. Results are averaged over 100 runs. Our analysis explains this phenomenon (see text in Section 6).

## B.1 Simulations varying dimension and the privacy parameter

In the simulations presented in Section 6, we fixed the privacy parameters to $\varepsilon = 1, \delta = 10^{-4}$ and the dimension to $d = 2$. We provide simulations that verify two expected behaviors that can be derived from the convergence rate provided in Theorem 4.1 when varying $\varepsilon$ and $d$. Namely, the accuracy degrades as the dimension $d$ grows, and improves when $\varepsilon$ grows.

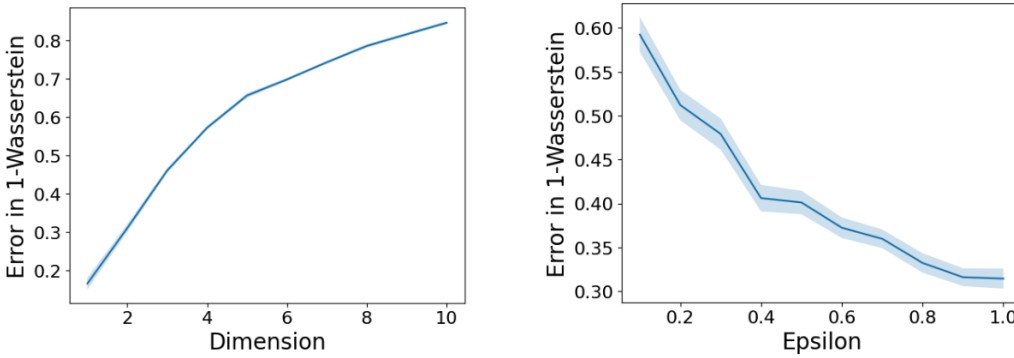

Figure 6: We set $\delta = 10^{-4}$ and run PE with the hyperparameter setting suggested by Theorem 4.1. For the plot on the right, we fixed $\varepsilon = 1$ and vary $d \in \{1, 2, 3, ..., 10\}$. For each value of $d$, we report the final accuracy of PE averaged over 100 runs. On the right, we repeat a similar procedure but fixing $d = 2$ and varying $\varepsilon \in \{0.1, 0.2, ..., 1\}$.

## C  Comparison between Algorithms 1, 2 and Practical PE

### C.1  Practical Private Evolution

Let us start introducing Algorithm 3, the practical implementation of the PE algorithm from [33].

### C.2  Comparison between Algorithm 1 and practical PE

We outline the main differences between the practical Algorithm 3 and the theoretical model in Algorithm 1. Algorithm 1 initializes $S_0$ with $n$ samples, and with high probability $|S_0| = |S_1| =$

---

**Algorithm 3** (Practical) Private Evolution [33]

---

1: **Input:** sensitive dataset: $S \in \Omega^n$, number of iterations: $T$, number of generated samples: $n_s$, noise multiplier: $\sigma$, distance function: $\rho(\cdot, \cdot)$, threshold $H > 0$
2: **Output:** DP synthetic dataset: $S_T \in \Omega^{n_s}$
3: $S_0 \leftarrow \text{Random\_API}(n_s)$
4: **for** $t = 1 \ldots T$ **do**
5: $\quad V_t \leftarrow \text{Variation\_API}(S_{t-1})$
6: $\quad \hat{\mu}_t \leftarrow \text{NN\_histogram}(S, V_t, \rho) \in \Delta_{|V_t|}$ (see Alg. 4)
7: $\quad \tilde{\mu}_t \leftarrow \hat{\mu}_t + \mathcal{N}(0, \sigma^2 I_{|V_t|})$
8: $\quad \mu'_t \leftarrow \mu'_t / \|\mu'_t\|_1$, where $\mu'_t[i] \leftarrow \tilde{\mu}_t[i] \mathbb{1}_{(\tilde{\mu}_t[i] \geq H)}$ for every $i \in [|V_t|]$
9: $\quad S_t \leftarrow n_s$ samples with replacement from $\mu'_t$
10: **end for**
11: **return** $S_T$

---

---

**Algorithm 4** Nearest Neighbor Histogram (NN\_histogram)

---

1: **Input:** datasets: $S, V \in \cup_{m \in \mathbb{N}} \Omega^m$, metric on $\Omega$: $\rho(\cdot, \cdot)$
2: **Output:** nearest neighbors histogram on $V$
3: histogram $\leftarrow (0, \ldots, 0) \in \mathbb{R}^{|V|}$
4: **for** $i = 1, \ldots, |S|$ **do**
5: $\quad k \leftarrow \min \left\{ k : k \in \arg\min_{j \in [|V|]} \rho(S[i], V[j]) \right\}$ (ties are broken by picking the smallest index in the argmin)
6: $\quad$ histogram$[k] \leftarrow$ histogram$[k] + 1/|S|$
7: **end for**
8: **return** histogram

---

$\ldots = |S_T|$. This is required for their utility proof based on $\eta$-closeness to work. In practice, PE maintains synthetic datasets of size $n_s$ potentially different from $n$, and $n_s$ is part of the input. In our simulations (section 6) we show that setting $n_s$ correctly is critical for convergence, hence not being able to set it differently from $n$ is a serious algorithmic limitation.

Once the NN is privatized by adding Gaussian noise and thresholded, Algorithm 1 creates the next dataset deterministically by adding variations to the next dataset proportionally to the entries of the DP histogram. There are two issues with this. First, calculating the proportions requires exact knowledge of the multiplicity parameter $B$ to create a dataset $\eta$-close to $S$. However, in practice $B$ should be calculated privately from $S$, which gives access to $B$ up to an error. Second, the practical version of PE normalizes the DP NN histogram after thresholding and then samples with replacement variations to create the next dataset. Algorithm 1 does not take into account these steps, and as a consequence its convergence analysis does not take into account the error incurred by them.

### C.3 More details on the limitations of the analysis of Algorithm 1 from [33]

The theoretical analysis of PE in [33] does not reflect how PE operates in practice. Their convergence proof relies on first showing that noiseless PE (Algorithm 1 with $\sigma = 0$) converges, then arguing that when each data point from the sensitive dataset $S$ repeated $B$ times, with high probability the noisy version behaves exactly the same for large $B$.

The convergence of noiseless PE is proven as follows. Given $S \in \Omega^n$ and the initial random $S_0 \in \Omega^n$, the elements of $S$ and $S_0$ are matched: assign $S_0[i]$ to $S[i]$. Variations of $S_0$, denoted by $V_0$, are created and $S_1[i]$ is chosen as the element from $V_0$ that is closest to $S[i]$, as indicated by the NN histogram. $V_0$ contains the variations of $S_0[i]$, which are likely to contain a point closer to $S[i]$ than $S_0[i]$. Repeating this process, they construct $S_0, S_1, \ldots, S_T$, where $S_t[i]$ is closer to $S[i]$ than $S_{t-1}[i]$. They conclude that for any $\eta > 0$ and sufficiently large $T$, it holds that for every $i \in [n]$, $S_T[i]$ is within distance $\eta$ of $S[i]$. We say that $S$ and $S_T$ are $\eta$-close when this happens, denoted by $S =_\eta S_T$.

To prevent noise from overwhelming the vote signal in the noisy case, [33] assumes that each data point in $S$ is repeated $B$ times. This ensures that, with high probability, the NN histogram entries are at least $B/n$ (dominating the added noise) and that the number of times $\text{nint}(n\mu'_t[i]/B)B$ that

$V_t[i]$ is included in $S_{t+1}$ is equal to $n\hat{\mu}_t[i]$. In other words, with high probability the noisy NN histogram coincides with the non-DP NN histogram, which clearly violates DP when we remove the unrealistic multiplicity assumption. In fact, the exact assumption they make to argue that $S =_\eta S_T$ is $B \gg \sqrt{d \log(1/\eta) \log(1/\delta)}/\varepsilon$ (Theorem 2 in [33]). If we let $B$ be a constant, then their convergence proof is valid only for $\varepsilon = \Omega(\sqrt{d \log(1/\eta) \log(1/\delta)})$ or, equivalently, $\eta = \Omega\left(e^{-\varepsilon^2/(d \log(1/\delta))}\right)$.

### C.4 Comparison between Algorithms 2 and 1

We note that Algorithm 2 overcomes the unrealistic features present in Algorithm 1. For example, it can generate random datasets of any number of samples $n_s$, since we do not use $\eta$-closeness arguments to prove its convergence. In addition, it does not require exact knowledge of the multiplicity parameter $B$, since our convergence proof assumes $B = 1$, which is usually the case in practice. Finally, Algorithm 2 post-processes the noisy NN histogram by projecting it to the spaces of probability measures over the variations and then samples variations according to this probability distribution to generate the next dataset, resembling what is done in practice.

## D  More Definitions

**Definition D.1** (Covering and packing number [45]). *Let $(\Omega, \rho)$ be a metric space. For $\delta > 0$, a $\delta$-cover of $\Omega$ w.r.t $\rho$ is a set $\Omega' \in \Omega^M$ such that for all $\omega \in \Omega, \exists i \in [M]$ with $\rho(\omega, \Omega'[i]) \leq \delta$. The covering number definition is the following:*

$$\mathcal{N}(\Omega, \rho; \delta) = \min\{n \in \mathbb{N} : \text{ exists a } \delta\text{-covering with } n \text{ elements}\}.$$

*Similarly, a $\delta$-packing of $\Omega$ w.r.t $\rho$ is a set $\Omega' \in \Omega^M$ with the property that $\forall(i,j) \in [M] \times [M]$ with $i \neq j$, $\rho(\Omega'[j], \Omega'[i]) \geq \delta$. The packing number is defined as*

$$\mathcal{M}(\Omega, \rho; \delta) = \max\{n \in \mathbb{N} : \text{ exists a } \delta\text{-packing with } n \text{ elements}\}.$$

**Definition D.2** (Gaussian and Laplace Complexity). *Let $\mathcal{F}$ be a function class on a metric space $(\Omega, \rho)$. Then, the Gaussian complexity [45] of $\mathcal{F}$ is defined as*

$$G_n(\mathcal{F}) := \sup_{z_1, \ldots, z_n \in \Omega} \mathbb{E}_{x_1, \ldots, x_n \overset{iid}{\sim} \mathcal{N}(0,1)} \left[ \sup_{f \in \mathcal{F}} \frac{1}{n} \left| \sum_{i \in [n]} x_i f(z_i) \right| \right].$$

*The Laplacian complexity [25] of $\mathcal{F}$ is defined as*

$$L_n(\mathcal{F}) := \sup_{z_1, \ldots, z_n \in \Omega} \mathbb{E}_{x_1, \ldots, x_n \overset{iid}{\sim} Lap(1)} \left[ \sup_{f \in \mathcal{F}} \frac{1}{n} \left| \sum_{i \in [n]} x_i f(z_i) \right| \right].$$

## E  Proof of main result

In this section we provide the proof of our main result, Theorem 4.1, along with the necessary auxiliary tools.

*Proof of Theorem 4.1.* Note that for any $t = 0, ..., T-1$

$$
\begin{aligned}
W_1(\mu_S, \mu_{S_{t+1}}) &= D_{BL}(\mu_S, \mu_{S_{t+1}}) \\
&\leq D_{BL}(\mu_S, \hat{\mu}_{t+1}) + D_{BL}(\hat{\mu}_{t+1}, \tilde{\mu}_{t+1}) + D_{BL}(\tilde{\mu}_{t+1}, \mu'_{t+1}) + D_{BL}(\mu'_{t+1}, \mu_{S_{t+1}}) \\
&= W_1(\mu_S, \hat{\mu}_{t+1}) + D_{BL}(\hat{\mu}_{t+1}, \tilde{\mu}_{t+1}) + D_{BL}(\tilde{\mu}_{t+1}, \mu'_{t+1}) + W_1(\mu'_{t+1}, \mu_{S_{t+1}}) \\
&\leq W_1(\mu_S, \hat{\mu}_{t+1}) + 2 D_{BL}(\hat{\mu}_{t+1}, \tilde{\mu}_{t+1}) + W_1(\mu'_{t+1}, \mu_{S_{t+1}}),
\end{aligned}
$$

where the inequality follows from $D_{BL}(\hat{\mu}_{t+1}, \tilde{\mu}_{t+1}) \geq D_{BL}(\tilde{\mu}_{t+1}, \mu'_{t+1})$ by definition of the projection in $D_{BL}$ distance. Denote by $\mathbb{E}_t[\cdot]$ the expectation when conditioning on the randomness up to iteration $t$ of PE. Lemmata E.1 and E.2 give

$$\mathbb{E}_t[W_1(\mu_S, \hat{\mu}_{t+1})] \leq (1-\gamma) W_1(\mu_S, \mu_{S_t}) + \alpha,$$

with $\gamma = \frac{1}{8\pi[(\sqrt{d}+\log(2))^2+\log(2)]}$. Further, using the definition of $\text{Variation\_API}(S_t)$, it is easy to see that $V_t$ has $n_s(2\lceil\log_2(D/\alpha)\rceil+1)$ elements. Letting $V = n_s(2\lceil\log_2(D/\alpha)\rceil+1)$, Lemma E.3 implies

$$\mathbb{E}_t[D_{BL}(\hat{\mu}_{t+1},\tilde{\mu}_{t+1})] \leq V\sigma G_V(\mathcal{F}_{BL}).$$

Putting everything together, we conclude that

$$\mathbb{E}_t[W_1(\mu_S,\mu_{S_{t+1}})] \leq (1-\gamma)W_1(\mu_S,\mu_{S_t}) + \alpha + 2V\sigma G_V(\mathcal{F}_{BL}) + W_1(\mu'_{t+1},\mu_{S_{t+1}}).$$

Integrating on both sides and denoting $\Gamma_t = \mathbb{E}[W_1(\mu_S,\mu_{S_t})]$ we obtain

$$\Gamma_{t+1} \leq (1-\gamma)\Gamma_t + \alpha + 2V\sigma\hat{G}_V(\mathcal{F}_{BL}) + \hat{W}_1(n_s),$$

where

$$\hat{G}_V(\mathcal{F}_{BL}) = 10D \begin{cases} \frac{\sqrt{\log(2\sqrt{V})}}{\sqrt{V}} & d = 1 \\ \frac{\log(2\sqrt{V})^{3/2}}{\sqrt{V}} & d = 2 \\ \frac{\sqrt{\log(2V^{1/d}(d/2-1)^{2/d})}}{V^{1/d}(d/2-1)^{2/d}} & d \geq 3 \end{cases}$$

is the upper bound on $\mathbb{E}[G_V(\mathcal{F}_{BL})]$ from Corollary E.1 and

$$\hat{W}_1(n_s) = CD \begin{cases} n_s^{-1/2} & d = 1 \\ \log(n_s)n_s^{-1/2} & d = 2 \\ n_s^{-1/d} & d \geq 3 \end{cases}$$

is the upper bound on $\mathbb{E}[W_1(\mu'_{t+1},\mu_{S_{t+1}})]$ from Lemma E.5. Hence, after $T$ steps of PE we obtain

$$\Gamma_T \leq (1-\gamma)^T\Gamma_0 + (\alpha + 2V\sigma\hat{G}_V(\mathcal{F}_{BL}) + \hat{W}_1(n_s))\sum_{i=0}^{T-1}(1-\gamma)^i$$

$$= (1-\gamma)^T\Gamma_0 + (\alpha + 2V\sigma\hat{G}_V(\mathcal{F}_{BL}) + \hat{W}_1(n_s))\frac{1-(1-\gamma)^T}{1-(1-\gamma)}$$

$$= (1-\gamma)^T\left[\Gamma_0 - \frac{\alpha + 2V\sigma\hat{G}_V(\mathcal{F}_{BL}) + \hat{W}_1(n_s)}{\gamma}\right] + \frac{\alpha + 2V\sigma\hat{G}_V(\mathcal{F}_{BL}) + \hat{W}_1(n_s)}{\gamma} \quad (2)$$

$$\leq e^{-\gamma T}D + \frac{\alpha + 2V\sigma\hat{G}_V(\mathcal{F}_{BL}) + \hat{W}_1(n_s)}{\gamma},$$

where the last inequality follows from the fact that $\frac{\alpha+2V\sigma\hat{G}_V(\mathcal{F}_{BL})+\hat{W}_1(n_s)}{\gamma} \geq 0$ and $\Gamma_0 \leq D$. Next, note that

$$2V\sigma G_V(\mathcal{F}_{BL}) + \hat{W}_1(n_s) = D \begin{cases} 20\sqrt{V}\sigma\sqrt{\log(2\sqrt{V})} + Cn_s^{-1/2} & d = 1 \\ 20\sqrt{V}\sigma\log(2\sqrt{V})^{3/2} + C\log(n_s)n_s^{-1/2} & d = 2 \\ \frac{20V^{1-1/d}\sigma\sqrt{\log(2V^{1/d}(d/2-1)^{2/d})}}{(d/2-1)^{2/d}} + Cn_s^{-1/d} & d \geq 3 \end{cases}.$$

Since $V = n_s(2\lceil\log_2(D/\alpha)\rceil+1)$, picking $n_s = \left(\sigma(2\lceil\log_2(D/\alpha)\rceil+1)^{1-1/\max\{d,2\}}\right)^{-1}$ gives

$$2V\sigma G_V(\mathcal{F}_{BL}) + \hat{W}_1(n_s) \leq \tilde{O}(D\sigma^{1/\max\{d,2\}}).$$

We conclude that

$$\Gamma_T \leq e^{-\gamma T}D + \tilde{O}\left(\frac{\alpha + D\sigma^{1/\max\{d,2\}}}{\gamma}\right).$$

Finally, setting $\alpha = D\sigma^{1/\max\{d,2\}}$ and $T \geq \lceil\log(\gamma/[D\sigma^{1/\max\{d,2\}}])/\gamma\rceil$ we obtain

$$\Gamma_T \leq \tilde{O}\left(\frac{D\sigma^{1/\max\{d,2\}}}{\gamma}\right) = \tilde{O}(dD\sigma^{1/d}).$$

This proves the first result. For the second, note that if $\varepsilon,\delta < 1$, then $\sigma = \frac{4\sqrt{T\log(1.25/\delta)}}{n\varepsilon}$ is enough noise to privatize $T$ iterations of PE by viewing it as the adaptive composition of

$T$ Gaussian mechanisms where each of them adds noise to a NN histogram with $\ell_2$ sensitivity $\sqrt{2}/n$ ([15], Corollary 3.3). Since we need $T \geq \log(\gamma/\sigma^{1/\max\{d,2\}})/\gamma$, it suffices to set $T = \frac{\log(n\varepsilon/[4\sqrt{\log(1/\delta)}])}{\max\{d,2\}\gamma} = O(\log(n\varepsilon/\sqrt{\log(1/\delta)}))$ to obtain

$$\Gamma_t \leq \tilde{O}\left(dD\left(\frac{\sqrt{\log(n\varepsilon/\sqrt{\log(1/\delta)})\log(1/\delta)}}{n\varepsilon}\right)^{1/d}\right).$$

$\square$

### E.1 Lemmata used in proof of Theorem 4.1

**Lemma E.1.** *Assume* $\Omega$ *is closed and convex. Then, for any* $z_1, z_2 \in \Omega$, $\mathbb{E}[\min_{z\in\text{Variation\_API}(z_1)}\|z-z_2\|_2] \leq (1-\gamma)\|z_1-z_2\|_2 + \alpha$ *with* $\gamma = \frac{1}{8\pi[(\sqrt{d}+\log(2))^2+\log(2)]}$, *where the expectation is taken w.r.t* $\text{Variation\_API}(z_1)$.

*Proof.*
- Case 1: $\|z_1-z_2\|_2 \leq \alpha$. Note that $\mathbb{E}[\min_{z\in\text{Variation\_API}(z_1)}\|z-z_2\|_2] \leq \|z_1-z_2\|_2 = \alpha$, since $z_1 \in \text{Variation\_API}(z_1)$.

- Case 2: $\|z_1-z_2\|_2 \geq \alpha$. In this case, there exists $l^* \in \{1,...,\lceil\log_2(\text{diam}(\Omega)/\alpha)\rceil\}$ such that $2^{l^*-1}\alpha \leq \|z_1-z_2\|_2 \leq 2^{l^*}\alpha$. Now, we proceed as follows,

$$\mathbb{E}\left[\min_{z\in\text{Variation\_API}(z_1)}\|z-z_2\|_2^2\right] = \mathbb{E}\left[\min_{k\in[2],l\in[L]}\|\text{proj}_\Omega(z_1+N_t^{k,l})-z_2\|^2\right]$$

$$\leq \mathbb{E}\left[\min_{k\in[2],l\in[L]}\|z_1+N_t^{k,l}-z_2\|^2\right]$$

$$= \mathbb{E}\left[\min_{k\in[2],l\in[L]}\|z_1-z_2\|_2^2 - 2\langle N_t^{k,l}, z_1-z_2\rangle + \|N_t^{k,l}\|^2\right]$$

$$\leq \min_{l\in[L]}\mathbb{E}\left[\min_{k\in[2]}\|z_1-z_2\|_2^2 - 2\langle N_t^{k,l}, z_1-z_2\rangle + \|N_t^{k,l}\|^2\right]$$

$$\leq \|z_1-z_2\|_2^2 + \min_{l\in[L]}\left\{-2\mathbb{E}\left[\max_{k\in[2]}\langle N_t^{k,l}, z_1-z_2\rangle\right] + \mathbb{E}\left[\max_{k\in[2]}\|N_t^{k,l}\|^2\right]\right\}$$

$$\leq \|z_1-z_2\|_2^2 + \min_{l\in[L]}\left\{-2\frac{\sigma_l\|z_1-z_2\|_2}{\sqrt{\pi}} + \sigma_l^2[(\sqrt{d}+\log(2))^2+\log(2)]\right\}$$

$$\leq \|z_1-z_2\|_2^2 - \frac{2\sigma_{l^*}2^{l^*-1}\alpha}{\sqrt{\pi}} + \sigma_{l^*}^2[(\sqrt{d}+\log(2))^2+\log(2)]$$

$$= \|z_1-z_2\|_2^2 - \frac{(\alpha 2^{l^*-1})^2}{\pi[(\sqrt{d}+\log(2))^2+\log(2)]}$$

$$\leq \|z_1-z_2\|_2^2\left(1 - \frac{1}{4\pi[(\sqrt{d}+\log(2))^2+\log(2)]}\right),$$

where we used the lower bound for the expected maximum of Gaussians from [29] and the upper bound for the expected maximum of chi-squared from [10] (Example 2.7), and the facts that $\sigma_l = \frac{\alpha 2^{l-1}}{\sqrt{\pi}[(\sqrt{d}+\log(2))^2+\log(2)]}$ and $2^{l^*-1}\alpha \leq \|z_1-z_2\|_2 \leq 2^{l^*}\alpha$. Jensen's inequality together with $\sqrt{1-t} \leq 1 - t/2$ for $t \leq 1$ allow us to conclude.

$\square$

**Lemma E.2.** *Let* $\Omega$ *be closed and convex and* $S, S' \in \cup_{m\geq 1}\Omega^m$ *be two datasets,* $V = \text{Variation\_API}(S')$ *and* $\hat{\mu} = \text{NN\_histogram}(S,V,\rho)$ *(see Algorithm 4). Suppose that for any* $z_1 \in \Omega, z_2 \in S$, $\mathbb{E}[\min_{z\in\text{Variation\_API}(z_1)}\rho(z,z_2)] \leq (1-\gamma)\rho(z_1,z_2)+\alpha$. *Then,*

$$\mathbb{E}[W_1(\mu_S,\hat{\mu})] \leq (1-\gamma)W_1(\mu_S,\mu_{S'})+\alpha.$$

*Proof.* The general idea is the following. We will exploit the fact that the 1-Wasserstein distance is defined as an the infimum of expected transport cost over couplings. First, we will bound $W_1(\mu_S, \hat{\mu})$ by constructing an specific coupling between $\mu_S, \hat{\mu}$ that comes from the NN histogram. Then, by using Lemma E.1 we will show that the resulting expression is upper bounded by the expected transport cost with respect to any coupling between $\mu_S$ and $\mu_{S'}$, up to an additive $\alpha$ term. By taking infimum over the couplings we will obtain the final expression.

Let $\hat{\pi} \in \Pi(\mu_S, \hat{\mu})$ be the coupling defined by

$$\hat{\pi}(S[i], V[j]) = \frac{\mathbb{1}\left(j = \min\left\{k : k \in \arg\min_{j \in [|V|]} \rho(S[i], V[j])\right\}\right)}{|S|},$$

for every $i \in [|S|], j \in [|V|]$. Let's verify that $\hat{\pi} \in \Pi(\mu_S, \hat{\mu})$. Clearly $\hat{\pi}(S[i], V[j]) \geq 0$. Furthemore,

$$\sum_{j \in [|V|]} \hat{\pi}(S[i], V[j]) = \frac{1}{|S|}$$

equals the mass that $\mu_S$ assigns to $S[i]$, and

$$\sum_{i \in [|S|]} \hat{\pi}(S[i], V[j]) = \sum_{j \in [|V|]} \frac{\mathbb{1}\left(j = \min\left\{k : k \in \arg\min_{j \in [|V|]} \rho(S[i], V[j])\right\}\right)}{|S|}$$

equals the mass that $\hat{\mu}$ assigns to $V[j]$, since $\hat{\mu}$ comes from a nearest neighbor histogram according to Algorithm 4 ($\hat{\mu} = \mathrm{NN\_histogram}(S, V, \rho)$). Hence, the marginals of $\hat{\pi}$ are $\mu_S$ and $\hat{\mu}$, which proves that $\hat{\pi} \in \Pi(\mu_S, \hat{\mu})$. Next, note that

$$
\begin{aligned}
\mathbb{E}[W_1(\mu_S, \hat{\mu}_{t+1})] &= \mathbb{E}\left[\inf_{\pi \in \Pi(\mu_S, \hat{\mu}_{t+1})} \int_{\Omega \times \Omega} \rho(z_1, z_2) d\pi(z_1, z_2)\right] \\
&= \mathbb{E}\left[\inf_{\pi \in \Pi(\mu_S, \hat{\mu}_{t+1})} \sum_{i \in [|S|]} \sum_{j \in [|V|]} \rho(S[i], V[j]) \pi(S[i], V[j])\right] \\
&\leq \inf_{\pi \in \Pi(\mu_S, \hat{\mu}_{t+1})} \mathbb{E}\left[\sum_{i \in [|S|]} \sum_{j \in [|V|]} \rho(S[i], V[j]) \pi(S[i], V[j])\right] \\
&\leq \mathbb{E}\left[\sum_{i \in [|S|]} \sum_{j \in [|V|]} \rho(S[i], V[j]) \hat{\pi}(S[i], V[j])\right] \\
&= \mathbb{E}\left[\sum_{i \in [|S|]} \sum_{j \in [|V|]} \rho(S[i], V[j]) \frac{\mathbb{1}\left(j = \min\left\{k : k \in \arg\min_{j \in [|V|]} \rho(S[i], V[j])\right\}\right)}{|S|}\right] \\
&= \mathbb{E}\left[\sum_{i} \frac{\min_{z \in V} \rho(S[i], z)}{|S|}\right].
\end{aligned}
$$

To continue the chain of inequalities, let us consider any coupling $\pi' \in \Pi(\mu_S, \mu_{S'})$. By definition it satisfies $\sum_{j \in [|S'|]} \pi'(S[i], S'[j]) = \frac{1}{|S|}$. Hence,

$$\mathbb{E}\left[\sum_i \frac{\min_{z \in V} \rho(S[i], z)}{|S|}\right] = \mathbb{E}\left[\sum_{i \in [|S|]} \left(\sum_{j \in [|S'|]} \pi'(S[i], S'[j])\right) \min_{z \in V} \rho(S[i], z)\right]$$

$$\leq \mathbb{E}\left[\sum_{i \in [|S|]} \sum_{j \in [|S'|]} \pi'(S[i], S'[j]) \min_{z \in \text{Variation\_API}(S'[j])} \rho(S[i], z)\right]$$

$$= \sum_{i \in [|S|]} \sum_{j \in [|S'|]} \pi'(S[i], S'[j]) \mathbb{E}\left[\min_{z \in \text{Variation\_API}(S'[j])} \rho(S[i], z)\right]$$

$$\leq \sum_{i \in [|S|]} \sum_{j \in [|S'|]} \pi'(S[i], S'[j]) \max\{(1 - \gamma)\rho(S[i], S'[j]), \alpha\}$$

$$\leq (1 - \gamma) \sum_{i \in [|S|]} \sum_{j \in [|S'|]} \pi'(S[i], S'[j])\rho(S[i], S'[j]) + \alpha$$

$$= (1 - \gamma) \int_{\Omega \times \Omega} \rho(z_1, z_2) d\pi'(z_1, z_2) + \alpha,$$

where the second inequality comes from the assumption that for any $z_1 \in \Omega, z_2 \in S$, $\mathbb{E}[\min_{z \in \text{Variation\_API}(z_1)} \rho(z, z_2)] \leq (1 - \gamma)\rho(z_1, z_2) + \alpha$. Putting the inequalities together, we conclude that for any $\pi' \in \Pi(\mu_S, \mu_{S'})$ it holds that

$$\mathbb{E}[W_1(\mu_S, \hat{\mu}_{t+1})] \leq (1 - \gamma) \int_{\Omega \times \Omega} \rho(z_1, z_2) d\pi'(z_1, z_2) + \alpha.$$

Taking infimum over the couplings on the right hand side finishes the proof. $\qquad \square$

**Lemma E.3.** *Let $V \in \Omega^n$. Let $\mu \in \Delta_n$ be a probability measure supported on $V$. Let $Z = (Z_1, ..., Z_n) \sim \mathcal{N}(0, \sigma^2 I_n)$. Then,*

$$\mathbb{E}_Z[D_{BL}(\mu, \mu + Z)] \leq n\sigma G_n(\mathcal{F}_{BL}).$$

*Proof.* By definition of $D_{BL}$, we have

$$D_{BL}(\mu, \mu + Z) = \sup_{f \in \mathcal{F}_{BL}} \int_\Omega f(d\mu - d(\mu + Z)) = \sup_{f \in \mathcal{F}} \sum_{i \in [n]} f(V[i])Z_i.$$

Hence,

$$\mathbb{E}[D_{BL}(\mu, \mu + Z)] = \mathbb{E}\left[\sup_{f \in \mathcal{F}} \sum_{i \in [n]} f(V[i])Z_i\right] = n\sigma \mathbb{E}\left[\sup_{f \in \mathcal{F}} \frac{1}{n} \sum_{i \in [n]} \frac{f(V[i])Z_i}{\sigma}\right] \leq n\sigma G_n(\mathcal{F}_{BL}).$$

$\qquad \square$

## E.2  Upper bounds

We need upper bounds on Gaussian complexity terms and on the convergence of empirical measures in $W_1$. Let us start showing an upper bound on the Gaussian complexity via Dudley chaining. This proof follows closely the proof of upper bounds for Laplacian Complexity given in [25].

**Lemma E.4** (Gaussian Complexity bound). *Let $(\Omega, \rho)$ be a metric space and $\mathcal{F}$ a set of functions on $\Omega$. Suppose $\text{diam}(\Omega) \leq D$. Then*

$$G_n(\mathcal{F}) \leq C \inf_{\alpha \geq 0} \left[\alpha + \int_\alpha^D \sqrt{\log(\mathcal{N}(\mathcal{F}, \|\cdot\|_\infty; \beta))} d\beta\right],$$

*where $C$ is an absolute constant.*

*Proof.* For each $j \in \mathbb{Z}$, define $\varepsilon_j = 2^{-j}$ and let $T_j$ be a $\varepsilon_j$-covering of $\mathcal{F}$, such that $|T_j| = N(\mathcal{F}, \|\cdot\|_\infty; \varepsilon_j)$ (i.e, the cardinality of the cover is the covering number w.r.t. $\|\cdot\|_\infty$). Fix $f \in \mathcal{F}$ and let $\pi_{j,f}$ be the closest function to $f$ in $T_j$. Since we are assuming that $\Omega$ has a bounded diameter, there exists $j_0 \in \mathbb{Z}$ such that for all $k \leq j_0$, $|T_k| = 1$. Then, for any set $\{z_i\}_{i=1}^n$,

$$\sup_{f \in \mathcal{F}} \frac{1}{n} \left| \sum_{i \in [n]} x_i f(z_i) \right| \leq \sup_{f \in \mathcal{F}} \frac{1}{n} \left| \sum_{i \in [n]} x_i [f(z_i) - \pi_{m,f}(z_i)] \right| + \sum_{k=j_0+1}^m \sup_{f \in \mathcal{F}} \frac{1}{n} \left| \sum_{i \in [n]} x_i [\pi_{k,f}(z_i) - \pi_{k-1,f}(z_i)] \right|,$$

for all $m \in \mathbb{Z}, m > j_0$.

We will give in-expectation bounds on the two terms on the right hand side. First, we trivially have

$$\mathbb{E}\left[\sup_{f \in \mathcal{F}} \frac{1}{n} \left| \sum_{i \in [n]} x_i [f(z_i) - \pi_{m,f}(z_i)] \right|\right] \leq \mathbb{E}\left[\frac{1}{n} \sum_{i \in [n]} |x_i| \varepsilon_m\right] \leq \varepsilon_m = 2^{-m}.$$

For the second term, note that

$$\frac{1}{n} |\pi_{k,f}(z_i) - \pi_{k-1,f}(z_i)| \leq \frac{1}{n} |\pi_{k,f}(z_i) - \pi_{k-1,f}(z_i)| + \frac{1}{n} |f(z_i) - f_{k-1}(z_i)|$$
$$\leq \frac{\varepsilon_k + \varepsilon_{k-1}}{n}$$
$$\leq \frac{3\varepsilon_k}{n}.$$

Hence, $\frac{1}{n} \sum_{i \in [n]} x_i [\pi_{k,f}(z_i) - \pi_{k-1,f}(z_i)]$ is a $(3\varepsilon_k/\sqrt{n})$-subGaussian random variable. We conclude that

$$\mathbb{E}\left[\sup_{f \in \mathcal{F}} \frac{1}{n} \left| \sum_{i \in [n]} x_i [\pi_{k,f}(z_i) - \pi_{k-1,f}(z_i)] \right|\right] \leq C \frac{\varepsilon_k \sqrt{\log(N(\mathcal{F}, \|\cdot\|_\infty; \varepsilon_k))}}{\sqrt{n}},$$

since the maximum is over at most $N(\mathcal{F}, \|\cdot\|_\infty; \varepsilon_k)^2$ random variables, each of which is $(3\varepsilon_k/\sqrt{n})$-subGaussian.

Putting everything together, we conclude that

$$\mathbb{E}\left[\sup_{f \in \mathcal{F}} \frac{1}{n} \left| \sum_{i \in [n]} x_i f(z_i) \right|\right] \leq C\left[2^{-m} + \sum_{k=j_0+1}^m \frac{\varepsilon_k \sqrt{\log(N(\mathcal{F}, \|\cdot\|_\infty; \varepsilon_k))}}{\sqrt{n}}\right]$$
$$\leq C' \inf_{\alpha \geq 0}\left[\alpha + \frac{1}{\sqrt{n}} \int_\alpha^\infty \sqrt{\log(N(\mathcal{F}, \|\cdot\|_\infty; \beta))} d\beta\right].$$

Finally, since $N(\mathcal{F}, \|\cdot\|_\infty; \beta) = 1$ for $\beta > D$, the integral above can be computed between $\alpha$ and $D$. This finishes the proof. $\square$

**Corollary E.1.** *Let $\Omega \subset \mathbb{R}^d$ and $\rho(\cdot, \cdot) = \|\cdot - \cdot\|_2$. If $\mathrm{diam}(\Omega) \leq D$, then*

$$G_n(\mathcal{F}_{BL}) \leq \begin{cases} \frac{9\sqrt{\log(2\sqrt{n})}D}{\sqrt{n}} & d = 1 \\ \frac{10D \log(2\sqrt{n})^{3/2}}{\sqrt{n}} & d = 2 \\ \frac{10D\sqrt{\log(2n^{1/d}(d/2-1)^{2/d})}}{n^{1/d}(d/2-1)^{2/d}} & d \geq 3 \end{cases}.$$

*Proof.* From [23] (Lemma 4.2), we know that

$$N(\mathcal{F}_L, \|\cdot\|_\infty; \beta) \leq \left(\frac{8}{\beta}\right)^{N(\Omega, \|\cdot\|_2; \beta/2L)},$$

where $\mathcal{F}_L$ is the class of $L$-Lipschitz functions mapping $\Omega$ to $[-1, 1]$. It is easy to see that $D\mathcal{F}_{1/D} = \mathcal{F}_{BL}$, since $\mathcal{F}_{BL}$ contains all the 1-Lipschitz functions mapping $\Omega$ to $[-D, D]$. Then, for any

$\beta \in [0, D]$, $\log(\mathcal{N}(\mathcal{F}_{BL}, \|\cdot\|_{\infty}; \beta))$ can be bounded as follows

$$\log(\mathcal{N}(\mathcal{F}_{BL}, \|\cdot\|_{\infty}; \beta)) = \log(\mathcal{N}(D\mathcal{F}_{1/D}, \|\cdot\|_{\infty}; \beta))$$
$$= \log(\mathcal{N}(\mathcal{F}_{1/D}, \|\cdot\|_{\infty}; \beta/D))$$
$$\leq \mathcal{N}(\Omega, \|\cdot\|_2; \beta/2) \log\left(\frac{8D}{\beta}\right)$$
$$\leq \mathcal{N}(DB_2, \|\cdot\|_2; \beta/2) \log\left(\frac{8D}{\beta}\right)$$
$$= \mathcal{N}(B_2, \|\cdot\|_2; \beta/2D) \log\left(\frac{8D}{\beta}\right)$$
$$\leq \left(\frac{4D}{\beta} + 1\right)^d \log\left(\frac{8D}{\beta}\right)$$
$$\leq \left(\frac{5D}{\beta}\right)^d \log\left(\frac{8D}{\beta}\right)$$

where the last inequality uses the fact that $\beta \in [0, D]$. Replacing this expression in the bound given by Lemma E.4 we get

$$G_n(\mathcal{F}_{BL}) \leq \inf_{\alpha \geq 0}\left(\alpha + \frac{1}{\sqrt{n}}\int_\alpha^D \left(\frac{5D}{\beta}\right)^{d/2}\sqrt{\log\left(\frac{8D}{\beta}\right)}d\beta\right)$$
$$\leq \inf_{\alpha \geq 0}\left(\alpha + \frac{\sqrt{\log(8D/\alpha)}}{\sqrt{n}}\int_\alpha^D \left(\frac{5D}{\beta}\right)^{d/2}d\beta\right)$$
$$= \inf_{\alpha \geq 0}\left(\alpha + \frac{\sqrt{\log(8D/\alpha)}(5D)^{d/2}}{\sqrt{n}}\int_\alpha^D \beta^{-d/2}d\beta\right).$$

We compute the integral for different values of $d$:

- $d = 1$. In this case,

$$\int_\alpha^D \beta^{-d/2}d\beta = 2(\sqrt{D} - \sqrt{\alpha}) \leq 2\sqrt{D}.$$

  It follows that

$$G_n(\mathcal{F}_{BL}) \leq \inf_{\alpha \geq 0}\left(\alpha + \frac{\sqrt{\log(8D/\alpha)}2\sqrt{5}D}{\sqrt{n}}\right) \leq 2\frac{\sqrt{\log(2\sqrt{n})}2\sqrt{5}D}{\sqrt{n}},$$

  where the last inequality comes from picking $\alpha = \frac{2\sqrt{5}D}{\sqrt{n}}$ in the infimum.

- $d = 2$. In this case,

$$\int_\alpha^D \beta^{-d/2}d\beta = \log(D) - \log(\alpha) = \log(D/\alpha) \leq \log(8D/\alpha).$$

  It follows that

$$G_n(\mathcal{F}_{BL}) \leq \inf_{\alpha \geq 0}\left(\alpha + \frac{\log(8D/\alpha)^{3/2}5D}{\sqrt{n}}\right) \leq 2\frac{\log(2\sqrt{n})^{3/2}5D}{\sqrt{n}},$$

  where the last inequality comes from picking $\alpha = \frac{5D}{\sqrt{n}}$ in the infimum.

- $d \geq 3$. In this case,

$$\int_\alpha^D \beta^{-d/2}d\beta = \frac{\alpha^{1-d/2} - D^{1-d/2}}{d/2 - 1} \leq \frac{\alpha^{1-d/2}}{d/2 - 1}.$$

It follows that

$$G_n(\mathcal{F}_{BL}) \leq \inf_{\alpha \geq 0} \left( \alpha + \frac{\sqrt{\log(8D/\alpha)}(5D)^{d/2}}{\sqrt{n}} \frac{\alpha^{1-d/2}}{d/2-1} \right)$$

$$\leq 2 \frac{\sqrt{\log(2n^{1/d}(d/2-1)^{2/d})}5D}{n^{1/d}(d/2-1)^{2/d}},$$

where the last inequality comes from picking $\alpha = \frac{5D}{n^{1/d}(d/2-1)^{2/d}}$ in the infimum.

$\square$

The convergence of empirical measures in 1-Wasserstein is a topic of interest on its own, that has been studied in the past. We obtain the following corollary from [31]. We note the results in there hold more generally for Banach Spaces.

**Lemma E.5** (Corollary of Theorem 3.1 in [31])**.** *Let $\mu$ be a probability measure with a finite discrete support on $\mathbb{R}^d$ contained in $\{x \in \mathbb{R}^d : \|x - y\|_2 \leq R\}$ for some $y \in \mathbb{R}^d$, $\mu_N$ the empirical distribution of $N$ iid samples from $\mu$. Then, there exists an absolute constant $C$, such that, for all $N \geq 1$,*

$$\mathbb{E}[W_1(\mu, \mu_N)] \leq CR \begin{cases} N^{-1/2} & d = 1 \\ \log(N)N^{-1/2} & d = 2 \\ N^{-1/d} & d \geq 3 \end{cases}.$$

## F   Simulations with histogram from Section 4.2

We follow the same simulation setup from Section 6. Namely, we consider the sample space to be the unit $\ell_2$ ball in $\mathbb{R}^2$. We use privacy parameters $\varepsilon = 1, \delta = 10^{-4}$. We set $T = 2\log(n\varepsilon)$, and the noise $\sigma$ is computed with the analytic Gaussian mechanism ([4], Theorem 8). The rest of the parameters are set as indicated in Theorem 4.1. We run Algorithm 2, with the only difference that we post-process noisy histograms truncating at $H = 0$ and then re-normalize. In Figure 7, we refer to this algorithm as 'PE'.

In Section 4.2, we introduced an alternative histogram that is built by adding Laplace noise (only) to the NN entries that are positive and then truncating the noisy entries that fall below $H = 2\log(1/\delta)/(n\varepsilon) + 1/n$, followed by a re-normalization step so that the noisy histogram induces a probability measure over the variations. Running Algorithm 2 with this histogram (instead of the one that adds Gaussian noise to all the entries of the NN histogram, thresholds at 0 and re-normalizes) is referred to as 'PE with Laplace noise+thresholding' in Figure 7.

We argued in Section 4.2 that this alternative DP histogram can work very well when the dataset is highly clustered, but it can also lead to vacuous utility guarantees in less favorable cases. We illustrate this in figure 7.

## G   Extension to Banach Spaces

We note that Algorithm 2 can operate on any Banach space $(\Omega, \rho)$ as long as it has access to adequate APIs. Below we define a property on the variation API that suffices to prove algorithm convergence.

**Definition G.1.** *Let $1 > \gamma > 0, v \in \mathbb{N}$ and $\alpha > 0$. A (randomized) variation API is a $(\gamma, v, \alpha)$-API for a dataset $S$ if*

- *For all $z \in \Omega$, $z \in \mathrm{Variation\_API}(z)$ and $|\mathrm{Variation\_API}(z)| \leq v$.*

- *For all $z_1 \in \Omega, z_2 \in S$ such that $\rho(z_1, z_2) > \alpha$,*

$$\mathbb{E}_{\mathrm{Variation\_API}(\cdot)} \left[ \min_{z \in \mathrm{Variation\_API}(z_1)} \rho(z, z_2) \right] \leq (1 - \gamma)\rho(z_1, z_2).$$

If Variation_API satisfies this definition, then we can prove the following result.

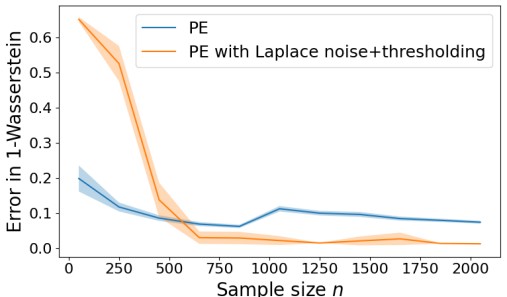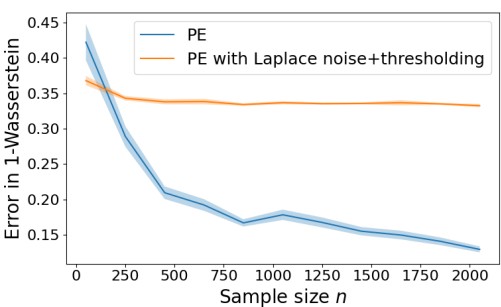

Figure 7: Comparison of the performance of PE with different histograms. Both algorithms, 'PE' and 'PE with Laplace noise+thresholding' are described in Section F. Left: the sensitive dataset is distributed uniformly in an $\ell_2$ ball of radious 0.02 around the origin (heavily clustered), leading 'PE with Laplace noise+thresholding' to preserve the votes signal in the noisy NN histograms, outperforming 'PE'. Right: the sensitive dataset is distributed uniformly in an $\ell_2$ ball of radious 0.5 around the origin (not clustered). The votes signal in the noisy NN histograms is lost by 'PE with Laplace noise+thresholding' during the thresholding step. Hence, it performs poorly. Results are averaged over 100 runs

**Theorem G.1** (Convergence of PE). *Let $(\Omega, \rho)$ be a (sample) Banach space. Suppose $\Omega$ is compact. Let $1 > \gamma > 0$, $v \in \mathbb{N}$ and $\alpha > 0$. $S_T$ be the output of Algorithm 2 run on input $S \in \Omega^n, T \in \mathbb{N}, n_s \in \mathbb{N}, \sigma > 0, \rho(\cdot, \cdot)$. If* $\text{Variation\_API}(\cdot)$ *is an $(\gamma, v, \alpha)$-API for $S$, then*

$$\mathbb{E}\left[W_1(\mu_S, \mu_{S_T})\right] \leq (1 - \gamma)^T \left[W_1(\mu_S, \mu_{S_0}) - err\right] + err,$$

*where $S_0$ is the dataset created in Line 3, $err = (\alpha + 2vn_s\sigma\hat{G}_{vn_s}(\mathcal{F}_{BL}) + \hat{W}_1(n_s))/\gamma$, $\hat{W}_1(n_s)$ is an upper bound on the rate of convergence of empirical measures from $n_s$ samples to the true measure in $W_1$ and $\hat{G}_{vn_s}(\mathcal{F}_{BL})$ is an upper bound on the Gaussian complexity $G_{vn_s}(\mathcal{F}_{BL})$.*

The upper bound on the Gaussian complexity that we provided in Lemma E.4 can be used to find $\hat{G}_{vn_s}(\mathcal{F}_{BL})$, since it works for general metric spaces, and the upper bound $\hat{W}_1(n_s))$, can be found in [31].

The proof of Theorem G.1 follows similarly to Theorem 4.1. We provide it below for completeness.

*Proof of Theorem G.1.* Note that for any $t = 0, ..., T - 1$

$$
\begin{aligned}
W_1(\mu_S, \mu_{S_{t+1}}) &= D_{BL}(\mu_S, \mu_{S_{t+1}}) \\
&\leq D_{BL}(\mu_S, \hat{\mu}_{t+1}) + D_{BL}(\hat{\mu}_{t+1}, \tilde{\mu}_{t+1}) + D_{BL}(\tilde{\mu}_{t+1}, \mu'_{t+1}) + D_{BL}(\mu'_{t+1}, \mu_{S_{t+1}}) \\
&= W_1(\mu_S, \hat{\mu}_{t+1}) + D_{BL}(\hat{\mu}_{t+1}, \tilde{\mu}_{t+1}) + D_{BL}(\tilde{\mu}_{t+1}, \mu'_{t+1}) + W_1(\mu'_{t+1}, \mu_{S_{t+1}}) \\
&\leq W_1(\mu_S, \hat{\mu}_{t+1}) + 2D_{BL}(\hat{\mu}_{t+1}, \tilde{\mu}_{t+1}) + W_1(\mu'_{t+1}, \mu_{S_{t+1}}),
\end{aligned}
$$

where the inequality follows from $D_{BL}(\hat{\mu}_{t+1}, \tilde{\mu}_{t+1}) \geq D_{BL}(\tilde{\mu}_{t+1}, \mu'_{t+1})$ by definition of $\mu'_{t+1}$. Denote by $\mathbb{E}_t[\cdot]$ the expectation when conditioning on the randomness up to iteration $t$ of PE. Since $\text{Variation\_API}(\cdot)$ is an $(\gamma, v, \alpha)$-API, then Lemma E.2 gives

$$\mathbb{E}_t[W_1(\mu_S, \hat{\mu}_{t+1})] \leq (1 - \gamma)W_1(\mu_S, \mu_{S_t}) + \alpha.$$

Further, since $|V_t| = vn_s$, Lemma E.3 and the definition of $\hat{G}_{n_s}(\mathcal{F}_{BL})$ imply

$$\mathbb{E}_t[D_{BL}(\hat{\mu}_{t+1}, \tilde{\mu}_{t+1})] \leq vn_s\sigma G_{vn_s}(\mathcal{F}_{BL}) \leq vn_s\sigma\hat{G}_{vn_s}(\mathcal{F}_{BL}).$$

Finally, by definition of $\hat{W}_1(n_s)$,

$$\mathbb{E}_t[W_1(\mu'_{t+1}, \mu_{S_{t+1}})] \leq \hat{W}_1(n_s).$$

Putting everything together:

$$\mathbb{E}_t[W_1(\mu_S, \hat{\mu}_{t+1})] \leq (1 - \gamma)W_1(\mu_S, \mu_{S_t}) + \alpha + 2n_s\sigma\hat{G}_{n_s}(\mathcal{F}_{BL}) + \hat{W}_1(n_s).$$

Integrating on both sides and denoting $\Gamma_t = \mathbb{E}[W_1(\mu_S, \mu_{S_t})]$ we obtain

$$\Gamma_{t+1} \leq (1 - \gamma)\Gamma_t + \alpha + 2vn_s\sigma\hat{G}_{vn_s}(\mathcal{F}_{BL}) + \hat{W}_1(n_s).$$

Hence, after $T$ steps of PE we obtain

$$\Gamma_T \leq (1 - \gamma)^T\Gamma_0 + (\alpha + 2vn_s\sigma\hat{G}_{vn_s}(\mathcal{F}_{BL}) + \hat{W}_1(n_s)) \sum_{i=0}^{T-1}(1 - \gamma)^i$$

$$= (1 - \gamma)^T\Gamma_0 + (\alpha + 2vn_s\sigma\hat{G}_{vn_s}(\mathcal{F}_{BL}) + \hat{W}_1(n_s))\frac{1 - (1 - \gamma)^T}{1 - (1 - \gamma)}$$

$$= (1 - \gamma)^T\left[\Gamma_0 - \frac{\alpha + 2vn_s\sigma\hat{G}_{vn_s}(\mathcal{F}_{BL}) + \hat{W}_1(n_s)}{\gamma}\right] + \frac{\alpha + 2vn_s\sigma\hat{G}_{vn_s}(\mathcal{F}_{BL}) + \hat{W}_1(n_s)}{\gamma}.$$

$\square$

# H   Missing proofs

We start with the proof of Proposition 4.1.

*Proof.* Recall that $\tilde{\mu}$ is given by $\tilde{\mu}[i] = (\hat{\mu}[i] + L_i)\mathbb{1}_{(\hat{\mu}[i]>0, \hat{\mu}[i]+L_i \geq H)}$ where $\{L_i\}_{i\in[m]} \overset{iid}{\sim}$ Lap$(2/n\varepsilon)$. This can be seen alternatively as constructing $\tilde{\mu}$ as follows

- If $\hat{\mu}[i] = 0$, then $\tilde{\mu}[i] = 0$.

- If $\hat{\mu}[i] > 0$, then $\tilde{\mu}[i] = \hat{\mu}[i] + L_i$, where $L_i \sim$ Lap$(2/n\varepsilon)$. If $\tilde{\mu}[i] < 2\log(1/\delta)/(n\varepsilon) + 1/n$ (i.e the coordinate is small), then it is truncated: $\tilde{\mu}[i] = 0$ .

Let $\nu$ be the measure after adding noise to $\hat{\mu}$ but before truncating the small coordinates to obtain $\tilde{\mu}$. Furthermore, note that all the entries with $\hat{\mu}[i] = 0$ remain unchanged with our procedure. Hence, the effective support of all the measures is $\hat{I} = \{i \in [m] : \hat{\mu}[i] > 0\}$ rather than $V$. Also, define $L(\beta)$ such that the event $E = \{|L_i| \leq L(\beta) \forall i\}$ has probability of at least $1 - \beta$. Recall that $H = 2\log(1/\delta)/(n\varepsilon) + 1/n$.

First, note that

$$W_1(\hat{\mu}, \mu') = D_{BL}(\hat{\mu}, \mu') \leq D_{BL}(\hat{\mu}, \nu) + D_{BL}(\nu, \tilde{\mu}) + D_{BL}(\tilde{\mu}, \mu').$$

We will control the terms on the right-hand side one by one.

- Let's start with $D_{BL}(\hat{\mu}, \nu)$.

$$\mathbb{E}[D_{BL}(\hat{\mu}, \nu)] = \mathbb{E}\left[\sup_{f\in\mathcal{F}_{BL}} \sum_{i\in\tilde{I}} f(V[i])(\hat{\mu}[i] + L_i - \hat{\mu}[i])\right]$$

$$= \frac{2|\tilde{I}|}{n\varepsilon}L_{|\tilde{I}|}(\mathcal{F}_{BL}),$$

where $L_{|\tilde{I}|}(\mathcal{F}_{BL})$ is the Laplace complexity of $\mathcal{F}_{BL}$.

- We continue by bounding $D_{BL}(\nu, \tilde{\mu})$.

$$\mathbb{E}[D_{BL}(\nu, \tilde{\mu})] = \mathbb{E}\left[\sup_{f \in \mathcal{F}_{BL}} \sum_{i \in \tilde{I}} f(V[i])(\hat{\mu}[i] + L_i - (\hat{\mu}[i] + L_i)\mathbb{1}_{(\hat{\mu}[i] + L_i \geq H)})\right]$$

$$\leq D\mathbb{E}\left[\sum_{i \in \tilde{I}:\hat{\mu}[i] + L_i < H} \hat{\mu}[i] + L_i\right]$$

$$\leq DB\mathbb{E}[|\{i \in \tilde{I} : \hat{\mu}[i] + L_i < H\}|]$$

$$\leq DB(\mathbb{E}[|\{i \in \tilde{I} : \hat{\mu}[i] + L_i < H\}| \mid E]\mathbb{P}[E] + \mathbb{E}[|\{i \in \tilde{I} : \hat{\mu}[i] + L_i < H\}| \mid E^C]\mathbb{P}[E^C])$$

$$\leq DB(|\{i \in \tilde{I} : \hat{\mu}[i] - L(\beta) \leq H\}| + |\tilde{I}|\beta)$$

- Finally, let's look at $D_{BL}(\tilde{\mu}, \mu')$. If $\|\tilde{\mu}\|_1 = 0$, then $D_{BL}(\tilde{\mu}, \mu') = 0$. Otherwise,

$$D_{BL}(\tilde{\mu}, \mu') = D_{BL}(\tilde{\mu}, \tilde{\mu}/\|\tilde{\mu}\|_1)$$

$$= \sup_{f \in \mathcal{F}_{BL}} \sum_{i \in \tilde{I}} f(V[i])\left(1 - \frac{1}{\|\tilde{\mu}\|_1}\right)\tilde{\mu}[i]$$

$$\leq D\left|1 - \frac{1}{\|\tilde{\mu}\|_1}\right|\|\tilde{\mu}\|_1 = D|\|\tilde{\mu}\|_1 - 1|.$$

Next,

$$|1 - \|\tilde{\mu}\|_1| = \left|\sum_{i \in \tilde{I}} \hat{\mu}[i] - \sum_{i \in \tilde{I}:\hat{\mu}[i] + L_i \geq H} \hat{\mu}[i] + L_i\right|$$

$$= \left|\sum_{i \in \tilde{I}:\hat{\mu}[i] + L_i < H} \hat{\mu}[i] + L_i - \sum_{i \in \tilde{I}} L_i\right|$$

$$\leq \left|\sum_{i \in \tilde{I}:\hat{\mu}[i] + L_i < H} \hat{\mu}[i] + L_i\right| + \left|\sum_{i \in \tilde{I}} L_i\right|$$

$$\leq B|\{i \in \tilde{I} : \hat{\mu}[i] + L_i < H\}| + \left|\sum_{i \in \tilde{I}} L_i\right|.$$

We have already argued that

$$\mathbb{E}[|\{i \in \tilde{I} : \hat{\mu}[i] + L_i < H\}|] \leq |\{i \in \tilde{I} : \hat{\mu}[i] - L(\beta) \leq H\}| + |\tilde{I}|\beta.$$

Furthemore, since the random variables $L_i$ are iid Laplace we have

$$\mathbb{E}\left[\left|\sum_{i \in \tilde{I}} L_i\right|\right] \leq \sqrt{\mathbb{E}\left[\left(\sum_{i \in \tilde{I}} L_i\right)^2\right]}$$

$$= \sqrt{Var\left(\sum_{i \in \tilde{I}} L_i\right)}$$

$$= \sqrt{|\tilde{I}|Var(L_1)}$$

$$= \sqrt{2|\tilde{I}|(2/(n\varepsilon))^2} = \frac{2\sqrt{2|\tilde{I}|}}{n\varepsilon}.$$

Hence, we conclude that

$$\mathbb{E}[D_{BL}(\tilde{\mu}, \mu')] \leq DH(|\{i \in \tilde{I} : \hat{\mu}[i] - L(\beta) \leq H\}| + |\tilde{I}|\beta) + \frac{2D\sqrt{2|\tilde{I}|}}{n\varepsilon}.$$

Putting everything together, we obtain

$$\mathbb{E}[W_1(\hat{\mu}, \mu')] \leq \frac{2|\tilde{I}|}{n\varepsilon} L_{|\tilde{I}|}(\mathcal{F}_{BL}) + 2DH(|\{i \in \tilde{I} : \hat{\mu}[i] - L(\beta) \leq H\}| + |\tilde{I}|\beta) + \frac{2D\sqrt{2|\tilde{I}|}}{n\varepsilon}.$$

Finally, observe that by the tail bounds of a Laplace random variable, $L(\beta) = O\left(\frac{\log(|\tilde{I}|/\beta)}{n\varepsilon}\right)$ suffices for $\mathbb{P}[E] \geq 1 - \beta$. This concludes the proof. $\qquad \square$

Next, we provide the proof of Lemma 3.1.

*Proof.* For simplicity denote the packing number $\mathcal{M}(\Omega, \rho; 2\eta)$ by $M$ in this proof. Consider a $2\eta$-packing of $\Omega$, $\{z_1, ..., z_M\}$. Let $S = \{z_1\}^n$. For $k \in \mathbb{Z}_+$, define
$$B(k) = B_\rho(z_1, \eta)^{n-1} \times B_\rho(z_k, \eta)$$
It is easy to see that if $S =_\eta \mathcal{A}(S) \iff \mathcal{A}(S) \in B(1)$, and if $\mathcal{A}(S) \in B(k)$ for some $k \geq 2$, then $S$ and $\mathcal{A}(S)$ can not be $\eta$-close. Since $\mathbb{P}_\mathcal{A}\left[S =_\eta \mathcal{A}(S)\right] \geq 1 - \tau$, then

$$\mathbb{P}_\mathcal{A}\left[\mathcal{A}(S) \in \cup_{k \in \mathbb{Z} \cap [2, M]} B(k)\right] \leq \tau.$$

Furthermore, $B(i) \cap B(j) = \emptyset$ for all $i \neq j \in \mathbb{Z} \cap [2, M]$, so

$$\mathbb{P}_\mathcal{A}\left[\mathcal{A}(S) \in \cup_{k \in \mathbb{Z} \cap [2, M]} B(k)\right] = \sum_{k \in \mathbb{Z} \cap [2, M]} \mathbb{P}_\mathcal{A}\left[\mathcal{A}(S) \in B(k)\right].$$

Hence, there exists a set $B(k^*)$ such with $\mathbb{P}_\mathcal{A}\left[\mathcal{A}(S) \in B(k^*)\right] \leq \tau/(M-1)$. Finally, construct $S' = \{z_1\}^{n-1} \cup \{z_{k^*}\}$. Note that $S$ and $S'$ are neighboring datasets. Then, by the inequalities that we have stated and $(\varepsilon, \delta)$-DP, it follows that

$$1 - \tau \leq \mathbb{P}_\mathcal{A}\left[S' =_\eta \mathcal{A}(S')\right] = \mathbb{P}_\mathcal{A}\left[\mathcal{A}(S') \in B(k^*)\right] \leq e^\varepsilon \mathbb{P}_\mathcal{A}\left[\mathcal{A}(S) \in B(k^*)\right] + \delta \leq \frac{e^\varepsilon \tau}{M-1} + \delta.$$

Solving for $\varepsilon$, we get $\varepsilon \geq \log((M-1)(1 - \tau - \delta)/\tau)$. Finally, note that
$$(M-1)(1 - \tau - \delta)/\tau \geq M(1 - \tau - \delta)/[2\tau] \geq M,$$
where the last inequality follows from the fact that $1 - \tau - \delta \geq 2\tau$, which is a consequence of our assumption that $\delta < 1 - 3\tau$. This concludes the proof. $\qquad \square$

Finally, we give a proof for Proposition 5.1.

*Proof.* Let $\mu \in \Delta_m$ and denote by $\Pi(\mu)$ is the set of couplings between $\frac{1}{n}\sum_{i \in [n]} \delta_{S[i]}$ and $\sum_{j \in [m]} \mu[j]\delta_{V[j]}$. Then, the following lower bound holds

$$W_1\left(\frac{1}{n}\sum_{i \in [n]} \delta_{S[i]}, \sum_{j \in [m]} \mu[j]\delta_{V[j]}\right) = \inf_{\pi \in \Pi(\mu)} \sum_i \left(\sum_j \rho(S[i], V[j])\pi_{ij}\right)$$

$$\geq \inf_{\pi \in \Pi(\mu)} \sum_i \left(\left(\min_{j \in [m]} \rho(S[i], V[j])\right) \sum_j \pi_{ij}\right)$$

$$= \sum_i \left(\frac{\min_{j \in [m]} \rho(S[i], V[j])}{n}\right). \tag{3}$$

In addition, note that if

$$\pi^*_{ij} = \frac{\mathbb{1}\left(j = \min\{k : k \in \arg\min_{l \in [m]} \rho(S[i], V[l])\}\right)}{n}$$

then (1) $\pi^* \in \Pi(\mu^*)$:

$$\sum_j \pi^*_{ij} = \frac{1}{n} \quad , \quad \sum_i \pi^*_{ij} = \mu^*[i],$$

and (2)

$$\sum_i \left( \sum_j \rho(S[i], V[j]) \pi^*_{ij} \right) = \sum_i \left( \sum_j \rho(S[i], V[j]) \pi^*_{ij} \right) = \sum_i \left( \frac{\min_{j \in [m]} \rho(S[i], V[j])}{n} \right).$$

(1) and (2) together with the lower bound from (3) imply that

$$(\mu^*, \pi^*) \in \arg \min_{\mu \in \Delta_m, \pi \in \Pi(\mu)} \sum_{i,j} \rho(S[i], V[j]) \pi_{ij},$$

which is equivalent to the statement we wanted to prove. $\qquad\square$

# I  Details on the Private Signed Measure Mechanism

The Private Signed Measure Mechanism is an algorithm for differentially private synthetic data generation presented in [25]. It is designed to work under pure differential privacy. For completeness, we provide an extension of it that works under approximate DP. The algorithm and its proof closely resemble the one from [25], with the only difference being that we need to deal with Gaussian noises instead of Laplace in order to achieve approximate DP. As we mentioned in Section 5, PE can be seen as a sequential version of PSMM, and the gaussian mechanisms has better composition guarantees than the Laplace mechanism. Hence, even though changing Gaussian noise by Laplace in PSMM does not make a big difference, it does make a more notorious difference for PE when seen as a sequential version of PSMM.

---

**Algorithm 5** Private signed measured mechanism with approx DP

---

**Require:** Dataset $S = \{z^1, ..., z^n\}$, partition $\{\Omega_i\}_{i \in [m]}$, number of synthetic samples $n_s$

1: Compute private counts: $\tilde{n}_i = |S \cap \Omega_i| + N_i$, where $N_i \overset{iid}{\sim} \mathcal{N}(0, \frac{\log(1/\delta)}{\varepsilon^2})$.
2: Let $\tilde{\mu}$ be a signed measure such that for $i \in [m]$, $\tilde{\mu}(\{\omega_i\}) = \tilde{n}_i/n$ for an arbitrary $\omega_i \in \Omega_i$ and $\tilde{\mu}(\Omega_i \setminus \{\omega_i\}) = 0$.
3: Let $\hat{\mu}$ be the closest probability measure over $\{\omega_1, ..., \omega_m\}$ to $\tilde{\mu}$ in $D_{BL}$ distance.
4: Let $\mu$ be an arbitrary probability measure over $\Omega$ such that $\mu(\Omega_i) = \hat{\mu}(\{\omega_i\})$ for all $i \in [m]$.
5: **return** $S' \sim \mu^{n_s}$

---

**Theorem I.1** (Guarantees of PSMM under approximate DP). *There exists a partition $\{\Omega_i\}_{i \in [m]}$ of $\Omega$ such that Algorithm 5 run on $S \in \Omega^n$, $\{\Omega_i\}_{i \in [m]}$ outputs an $(\varepsilon, \delta)$-DP synthetic dataset $S'$ satisfying*

$$\mathbb{E}[W_1(\mu_S, \mu_{S'})] \le 2 \left( \max_{i \in [m]} \mathrm{diam}(\Omega_i) + \frac{m\sqrt{\log(1/\delta)}}{n\varepsilon} G_m(\mathcal{F}) \right) + \mathbb{E}[W_1(\mu, \mu_{S'})],$$

*where $\mu$ is the probability measure from Step 4. Furthermore, if $\Omega \subset \mathbb{R}^d$ with $\mathrm{diam}(\Omega) \le D$, $\rho = \|\cdot\|_2$, $m = n\varepsilon/[D\sqrt{\log(1/\delta)}]$ and $\mathrm{diam}(\Omega_i) \le O(Dm^{-1/\max\{d,2\}})$, then*

$$\mathbb{E}[W_1(\mu_S, \mu)] \le \tilde{O} \left( D \left( \frac{\sqrt{\log(1/\delta)}}{n\varepsilon} \right)^{1/\max\{2,d\}} + D \left( \frac{1}{n_s} \right)^{1/\max\{2,d\}} \right).$$

**Remark I.1.** *When running PSMM, we can select $n_s$ arbitrarily large, by the post-processing property of DP. Hence, we can safely assume that the error term $D \left( \frac{\sqrt{\log(1/\delta)}}{n\varepsilon} \right)^{1/\max\{2,d\}}$ dominates in the utility bound presented in Theorem I.1. Note that, up to logarithmic factors, this bound improves over the bound that we provide for PE in Theorem 4.1 by a factor of $d$. However, it requires $\{\Omega_i\}_{i \in [m]}$ to be such that $\max_{i \in [m]} \mathrm{diam}(\Omega_i) = O(Dm^{-1/\max\{d,2\}})$. Constructing such partition in practice is equivalent to finding an $O(Dm^{-1/\max\{d,2\}})$-net of points $\{\omega_1, ...., \omega_m\}$ such that the Voronoi partition induced by them is $\{\Omega_i\}_{i \in [m]}$. We explained in Section 5 why such net can be difficult to construct in practice.*

*Proof.* Since the $\ell_2$-sensitivity of $f(S) = (|S \cap \Omega_i|)_{i \in [m]]}$ is $\sqrt{2}$, privacy of the measure $\mu$ (from Step 4) follows from the Gaussian mechanism. The privacy of $S'$ follows by post-processing.

The convergence proof is similar to the utility analysis of one step of PE. Let $\bar{\mu}$ be a probability measure over $\Omega$ such that $\bar{\mu}(\{\omega_i\}) = n_i/n$, where $n_i = |S \cap \Omega_i|$, and $\bar{\mu}(\Omega \backslash \{\omega_1, ..., \omega_m\}) = 0$. Let the measures $\tilde{\mu}, \hat{\mu}$ and $\mu$ be as given by Algorithm 5. Note that

$$
\begin{aligned}
W_1(\mu_S, \mu) = D_{BL}(\mu_S, \mu) &\leq D_{BL}(\mu_S, \bar{\mu}) + D_{BL}(\bar{\mu}, \tilde{\mu}) + D_{BL}(\tilde{\mu}, \hat{\mu}) + D_{BL}(\hat{\mu}, \mu) \\
&= W_1(\mu_S, \bar{\mu}) + D_{BL}(\bar{\mu}, \tilde{\mu}) + D_{BL}(\tilde{\mu}, \hat{\mu}) + W_1(\hat{\mu}, \mu) \\
&\leq W_1(\mu_S, \bar{\mu}) + 2D_{BL}(\bar{\mu}, \tilde{\mu}) + W_1(\hat{\mu}, \mu) \\
&\leq 2 \left( \max_{i \in [m]} \operatorname{diam}(\Omega_i) + D_{BL}(\bar{\mu}, \tilde{\mu}) \right),
\end{aligned}
$$

where the second inequality follows from the fact that $D_{BL}(\bar{\mu}, \tilde{\mu}) \geq D_{BL}(\tilde{\mu}, \hat{\mu})$, which is a consequence of $\hat{\mu}$ being the closest probability measure over $\{\omega_1, ..., \omega_m\}$ to $\tilde{\mu}$ in $D_{BL}$, and in the last inequality we used $\max\{W_1(\mu_S, \bar{\mu}), W_1(\hat{\mu}, \mu)\} \leq \max_{i \in [m]} \operatorname{diam}(\Omega_i)$ (these inequalities are trivial since the pairs of measures $\mu_S, \bar{\mu}$ and $\hat{\mu}, \mu$ assign the same amount of probability mass into each region $\Omega_i$). It remains to bound $D_{BL}(\bar{\mu}, \hat{\mu})$. The following equality

$$
D_{BL}(\bar{\mu}, \tilde{\mu}) = \sup_{f \in \mathcal{F}} \int f(d\bar{\mu} - d\tilde{\mu}) = \sup_{f \in \mathcal{F}} \sum_{i \in [m]} f(\omega_i) \left( \frac{n_i}{n} - \frac{n_i + N_i}{n} \right) = \sup_{f \in \mathcal{F}} \sum_{i \in [m]} \frac{f(\omega_i) N_i}{n}
$$

allows to conclude that

$$
\mathbb{E}[W_1(\mu_S, \mu)] \leq 2 \left( \max_{i \in [m]} \operatorname{diam}(\Omega_i) + \frac{m\sqrt{\log(1/\delta)}}{n\varepsilon} G_m(\mathcal{F}) \right).
$$

The proof of the first claim is concluded by noting that

$$
\begin{aligned}
\mathbb{E}[W_1(\mu_S, \mu_{S'})] &\leq \mathbb{E}[W_1(\mu_S, \mu)] + \mathbb{E}[W_1(\mu, \mu_{S'})] \\
&\leq 2 \left( \max_{i \in [m]} \operatorname{diam}(\Omega_i) + \frac{m\sqrt{\log(1/\delta)}}{n\varepsilon} G_m(\mathcal{F}) \right) + \mathbb{E}[W_1(\mu, \mu_{S'})].
\end{aligned}
$$

For the second claim, recall from Corollary E.1 that

$$
G_m(\mathcal{F}_{BL}) \leq \begin{cases} \frac{9\sqrt{\log(2\sqrt{m})}D}{\sqrt{m}} & d = 1 \\ \frac{10D \log(2\sqrt{m})^{3/2}}{\sqrt{m}} & d = 2 \\ \frac{10D\sqrt{\log(2m^{1/d}(d/2-1)^{2/d})}}{m^{1/d}(d/2-1)^{2/d}} & d \geq 3 \end{cases}.
$$

Using this in the inequality from the first claim and using that $\operatorname{diam}(\Omega_i) = O(Dm^{-1/\max\{2,d\}})$ for all $i \in [m]$ we obtain that

$$
\mathbb{E}[W_1(\mu_S, \mu_{S'})] = \tilde{O} \left( Dm^{-1/\max\{2,d\}} + \frac{Dm^{1-1/\max\{2,d\}}\sqrt{\log(1/\delta)}}{n\varepsilon} \right) + \mathbb{E}[W_1(\mu, \mu_{S'})].
$$

Using the definition $m = n\varepsilon/[D\sqrt{\log(1/\delta)}]$, it follows that

$$
\mathbb{E}[W_1(\mu_S, \mu_{S'})] = \tilde{O} \left( D \left( \frac{\sqrt{\log(1/\delta)}}{n\varepsilon} \right)^{1/\max\{2,d\}} \right) + \mathbb{E}[W_1(\mu, \mu_{S'})].
$$

Finally, since $S' \sim \mu^{n_s}$, Lemma E.5 gives the following upper bound on the convergence of the empirical measure in $W_1$

$$
\mathbb{E}[W_1(\mu, \mu_{S'})] = O(Dn_s^{-1/\max\{2,d\}}).
$$

This finishes the proof. $\qquad \square$

## J Impact Statement

Our work is theoretical in nature, so it does not have direct societal impacts. However, we believe that our theory can improve the practice of differentially private synthetic data generation, leading to a positive social impact by enabling safe data sharing within and across organizations.

