# OpenReview forum: "Private Evolution Converges"
_NeurIPS.cc/2025/Conference — NeurIPS 2025 poster_

### Official Review · Reviewer_gGaX · 2025-06-13

**Clarity:** 2
**Significance:** 2
**Originality:** 3
**Rating:** 3
**Confidence:** 5

**Summary:**

This paper presents a new theoretical analysis of Private Evolution (PE), a training-free method for differentially private synthetic data generation. The authors identify limitations in prior work and propose a variant of PE for theoretical study (Algorithm 2) that more closely resembles the original PE. The main contribution is an upper bound on the Wasserstein distance between the synthetic data and the private data. The paper also draws an interesting connection between PE and the Private Signed Measure Mechanism (PSMM).

**Questions:**

1. How does Lemma 3.1 change if we relax from pure DP to approximate DP?
2. What does "worst-case" refer to in the convergence guarantee?

**Ethical Concerns:**

["NO or VERY MINOR ethics concerns only"]

**Final Justification:**

I think it is very risky to have this paper accepted, as the central message that it delivers (PE converges) is contradictory to many practical evidence on PE's underperformance. Without clarifying where the gap lies in, I am not comfortable about recommending acceptance.

**Limitations:**

No, the authors didn't discuss the limitations of this work. The authors should discuss the mismatch between the theoretical results and the empirical evidence, and what are the potential causes.

**Quality:**

2

**Strengths And Weaknesses:**

**Strengths**:
- The theoretical analysis is sound and novel
- The connection between PE and PSMM is interesting. It offers a new perspective on PE as an iterative method for constructing discrete support for data release.

**Weaknesses**:
- **Misleading convergence claim.**  The title "Private Evolution Converges" is misleading. Typically, the convergence of an iterative algorithm implies that a quality metric improves as the number of iterations $T \to \infty$. The analysis in this paper does not show this. Instead, the paper establishes convergence only in the limit of the dataset size $n \to \infty$. Furthermore, for the expected 1-Wasserstein distance to approach a small constant, $n$ must scale as $O(d^d)$, which is impossible in practice.
- **Disconnect with empirical evidence.**  A major concern is the apparent disconnect between the paper's theoretical guarantees and the extensive empirical evidence on PE's behavior.
  - Notably, multiple studies have documented that PE suffers from a lack of diversity or **mode collapse** (see Figs. 21 and 22 in [1] and Fig. 4 in [2]). This makes distributional convergence impossible. This critical point is not captured by the current analysis and has not been discussed in this paper.
  - The analysis in Line 314 implies a monotonic progression of the Wasserstein distance, suggesting the optimal $T$ is either 1 or $\infty$. This does not capture the U-shaped performance curves (e.g., Fig. 4 of [2]) where the fidelity of the synthetic data (e.g., FID) first improves and then degrades (due to overfitting to majority modes).
- **Limited practical insights.**  The theoretical framework, while rigorous, struggles to provide new, actionable insights into PE's practical challenges. For example, the theory suggests that the optimal $T$ is 1 when the initial error is smaller than the asymptotic error term, which the authors use to explain PE's poor performance on tabular datasets. However, this explanation is not convincing, as the asymptotic error vanishes with $n$, and it is unclear why this condition would uniquely apply to tabular data.

In summary, while the paper succeeds as a theoretical exercise in deriving a bound for a variant of PE, it falls short of its goal to "explain PE's practical behavior". The core research question (convergence) is misaligned with PE's most pressing practical challenges, particularly mode collapse. A more interesting direction is to develop theory that explains why and when PE fails, as this would be more likely to guide the development of better algorithms for DP synthetic data generation (please cite [3] for this broad research topic).

**References**

[1] Lin, Zinan, et al. "Differentially private synthetic data via foundation model apis 1: Images." arXiv preprint arXiv:2305.15560 (2023).

[2] Hou, Charlie, et al. "Private Federated Learning using Preference-Optimized Synthetic Data." arXiv preprint arXiv:2504.16438 (2025).

[3] Hu, Yuzheng, et al. "Sok: Privacy-preserving data synthesis." 2024 IEEE Symposium on Security and Privacy (SP). IEEE, 2024.

---

> ### Author Rebuttal · Authors · 2025-07-31
>
> Thank you for your feedback. We address your concerns below, and we will include reference [3] in the related work section of our paper.
>
> >`Misleading convergence claim.`
>
> We agree that 'PE converges' could be interpreted as convergence as $T \to \infty$. This is why we clarified this point immediately in the abstract to indicate that it converges as $n\to \infty$, and in the main theorem we say that 'there exists a parameter setting' such that PE converges at the stated rate. In the DP literature, it is common to study convergence rates as a function of the number of samples. For example, the convergence rates of iterative algorithms for DP stochastic optimization are usually a function of the number of samples, ambient dimension and privacy and problem parameters, but usually they fix a nontrivial number of iterations in these methods that achieves optimal trade-offs between privacy and accuracy (e.g., SGD and Stochastic Frank-Wolfe variants in [6,7]); this is similar to what we do in our work. We will clarify this point in the paper.
>
> >`for the expected 1-Wasserstein distance to approach a small constant, $n$ must scale as $O(d^d), which is impossible in practice.`
>
> It has been proven that under DP constraints, any computationally efficient algorithm cannot accurately preserve all marginals [4] and W_1 not only preserves all marginals, it preserves all Lipschitz queries. There may be metrics that do not suffer from the curse of dimensionality, but for W_1 this price is inherent and not an issue specific to our proposed variant of PE. Further, $d$ could actually be the intrinsic dimensionality of the data (e.g. if the data lies on a low-dimensional manifold, the RANDOM_API and VARIATION_API may generate data only on that manifold). We will clarify these points in the paper.
>
> To improve the sample complexity, an interesting future direction would be to exploit more structure of specific PE instances. We proposed an analysis beyond the worst-case for very clustered data, but there are many other structural properties that one can exploit to improve the analysis, such as prior knowledge on the data and properties of the APIs.
>
> >`Disconnect with empirical evidence. (...) multiple studies have documented that PE suffers from a lack of diversity or mode collapse (see Figs. 21 and 22 in [1] and Fig. 4 in [2]). This makes distributional convergence impossible. This critical point is not captured by the current analysis and has not been discussed in this paper.`
>
> This is a good point; actually, our paper shows that the mode collapse problem is not inherent to PE; rather, one can get *distributional* convergence in terms of W_1 with sufficient samples (and our choice of APIs). In practice, mode collapse could arise from several sources that reduce diversity, such as low-quality APIs or insufficient private data. In fact, under privacy constraints we should not expect a DP algorithm to capture small clusters of datapoints; this may manifest as dropped modes. Supporting this view, in the references that you mention, the authors acknowledge that mode collapse can be reduced by improving diversity of the generated data, e.g., by running a VARIATION_API as a postprocessing step (Appendix L in [1]).
>
> We have run an experiment to illustrate our point. We show that for datasets with $4$ clusters of $n/4$ datapoints in $R^2$, our variant of PE struggles to capture the 4 modes (clusters) when $n=100$, but gradually improves as $n$ grows and captures the cluster well when $n=1000$ and extremely well when $n=10000$. We will include this experiment in our paper.
>
> `The analysis in Line 314 implies a monotonic progression of the Wasserstein distance, suggesting the optimal $T$ is either $1$ or $\infty$. This does not capture the U-shaped performance curves (e.g., Fig. 4 of [2]) where the fidelity of the synthetic data (e.g., FID) first improves and then degrades (due to overfitting to majority modes).`
>
> The equation in line 314 does not imply that the optimal $T$ is $1$ or $\infty$. We understand the confusion and will make the following more clear in the paper: the error term 'err' scales polynomially with the noise $\sigma$, which by composition of DP scales at least as $\sqrt{T}$. Hence, we do recover a U-shaped graph when plotting the error in $W_1$ as a function of $T$, as shown in Figure 3 of the paper. The point of our claim in line 314 is that it is possible that when the error term is large in early iterations, the optimal number of steps might be 0 or 1, as illustrated in our experiments in Section 6. In other cases the optimal number $T$ is not infinity, it is the solution to a nontrivial trade-off involving $T$ and other problem parameters (such as number of synthetic samples, privacy parameters, etc) that needs to be solved to maximize the accuracy. Finally, it is not clear to us that every time the error starts to degrade while running PE is because of overfitting to majority modes.
>
> `Limited practical insights. (...) the theory suggests that the optimal $T$ is 1 when the initial error is smaller than the asymptotic error term, which the authors use to explain PE's poor performance on tabular datasets. However, this explanation is not convincing, as the asymptotic error vanishes with $n$, and it is unclear why this condition would uniquely apply to tabular data.`
>
> As mentioned before, the correct way of thinking of the error term is as an expression that increases polynomially with $T$, because when running PE $n$ is fixed and hence it is not relevant that the error vanishes as $n \to \infty$. We will emphasize this in the paper.
>
> We believe that our explanation is feasible for the following reasons.
>
> 1. The definition of accuracy in [5] aligns well with the definition of accuracy used in our work: both works define error as a maximum discrepancy over a class of queries (as opposed to other works that measure error with downstream tasks or by embedding the data and calculating FID, which is a 2-Wasserstein distance), so it makes sense that their measure of error and ours behave similarly.
> 2. The findings of the authors in [5] align with our theory. Indeed, they claim that 'The observation that the workload-aware private evolution algorithm performs best with one shot data generation implies that: whatever marginal gains we get from iterating multiple times, they are outweighed by the privacy cost of composing over iterations'. This finding aligns well with our theoretical findings in the case where the error term 'err' from line 314 increases significantly after 1 iteration.
> 3. Finally, we do not claim that our observation applies uniquely to tabular data; it is straightforward to find PE runs in other domains where the optimal number of PE steps is also 1.
>
> `How does Lemma 3.1 change if we relax from pure DP to approximate DP?`
>
> Letting $M$ be the $2\eta$ packing number of $\Omega$, the lower bound of epsilon in Lemma 3.1 changes to from $\log((M-1)(1-\tau)/\tau)$ to $\log((M-1)(1-\tau-\delta)/\tau)$ for $\delta < 1-\tau$. This follows from including the $\delta$ term in the equation below line 1048. We will add this to the paper.
>
> `What does "worst-case" refer to in the convergence guarantee?`
>
> 'Worst-case' is over problem instances: for any dataset $S$ and dimension $d$, our proposed PE variant privately generates synthetic data whose empirical distribution is close to that of S with W_1 guarantees. It's not worse case over variants of PE though, we will add a remark on this.
>
> `the authors didn't discuss the limitations of this work. The authors should discuss the mismatch between the theoretical results and the empirical evidence, and what are the potential causes.`
>
> We will make explicit the limitations of our work, and clarify that we are not claiming that that the theory and the practice of PE are completely bridged. We will also discuss the potential causes, which are related to our choice of APIs and to the fact that in practice it is possible to exploit problem-dependent structure.
>
> References
>
>
> [1] Lin, Zinan, et al. "Differentially private synthetic data via foundation model apis 1: Images." arXiv preprint arXiv:2305.15560 (2023).
>
> [2] Hou, Charlie, et al. "Private Federated Learning using Preference-Optimized Synthetic Data." arXiv preprint arXiv:2504.16438 (2025).
>
> [3] Hu, Yuzheng, et al. "Sok: Privacy-preserving data synthesis." 2024 IEEE Symposium on Security and Privacy (SP). IEEE, 2024.
>
> [4] Ullman, Jonathan, and Salil Vadhan. "PCPs and the hardness of generating private synthetic data." Theory of Cryptography Conference.
>
> [5] M. Swanberg, R. McKenna, E. Roth, A. Cheu, and P. Kairouz. Is API access to LLMs useful for generating private synthetic tabular data? In ICLR Workshop: Will Synthetic Data Finally Solve the Data Access Problem?, 2025.
>
> [6] Bassily, Raef, et al. "Private stochastic convex optimization with optimal rates." Advances in neural information processing systems 32 (2019).
>
> [7] Bassily, Raef, Cristóbal Guzmán, and Anupama Nandi. "Non-euclidean differentially private stochastic convex optimization." Conference on Learning Theory (2021).

---

> ### Comment · Reviewer_gGaX · 2025-08-01
>
> Thank you for the response. I appreciate the clarification regarding the optimal choice of $T$.
>
> That said, my main concern remains. I worry that a general reader may walk away with a partial and in some sense even misleading message from this work. While PE is an elegant idea, in practice it is not a competitive algorithm for generating DP synthetic data. This is supported by several evidence in the literature: in addition to the works I previously mentioned, Tan et al. (2025) show that in the non-private setting, PE's utility significantly lags behind that of fine-tuning. In fact, its performance barely improves when moving from the DP to the non-DP regime. Evidence from industry also tells the same story: a recent ICML 2025 tutorial (https://icml.cc/virtual/2025/40009, registration required) highlights that PE has a very narrow application scope and is only usable in low data or low privacy budget regime.
>
> Collectively, these findings suggest that PE, and even "non-private" evolution, tends not to converge in practical settings. Yet this paper proposes a variant of PE that is claimed to “more closely reflect how PE works in practice”, and demonstrates that it converges. Without identifying the source of this discrepancy, the paper risks confusing researchers / practitioners already familiar with PE and potentially misleading newcomers to the field. Is the discrepancy due to insufficient private data? Limitations or issues in the variation_api (and if so, what are they)? Or perhaps the choice of embedding model? Since the paper makes a convergence claim that runs counter to widespread practitioner experience, I believe it is necessary for the authors to address this gap before the paper can be accepted. In my opinion, this will also make the work much stronger as it offers practitioners a clear guidance on how to improve PE.
>
> Lastly, like most other reviewers, I think the experiments on low-dimensional synthetic datasets are insufficient. While I appreciate the newly added experiments exploring the effect of increasing $n$ on synthetic data, I am more interested in understanding whether these findings generalize to real-world datasets. Specifically, does access to more private samples meaningfully improve the quality of synthetic data generated from PE?
>
> **References**
>
> Tan, Bowen, et al. "Synthesizing privacy-preserving text data via finetuning without finetuning billion-scale llms." arXiv preprint arXiv:2503.12347 (2025).

---

### Official Review · Reviewer_SoiT · 2025-06-26

**Clarity:** 3
**Significance:** 3
**Originality:** 3
**Rating:** 5
**Confidence:** 3

**Summary:**

Private Evolution (PE) is a synthetic data generation process that has shown some promising empirical results. A recent paper ([31] in the submission) gave convergence guarantees for a variant of PE, but this paper argues that the assumptions in [31] are quite unrealistic. Without those assumptions, the original method offers weak privacy guarantees. The current submission proposes a different (still not exactly the same as practical) variant of PE and proves worst-case convergence bounds for it.

**Questions:**

I’m a little confused about the implication of Lemma 3.1. There is no qualifier on $\mathcal{A}$. Does this mean the lower bound applies to *any* algorithm $\mathcal{A}$? If so, why doesn’t it apply to your algorithm? Or maybe you meant $\mathcal{A}$ refers specifically to Algorithm 1?

Also, just a minor formatting thing, but in Algorithm 1, the line $S_0 \gets \mathrm{Random} \\_ \mathrm{API}(n)$ is highlighted in blue, while the same step in Algorithm 2 isn’t. Was that intentional?

**Ethical Concerns:**

["NO or VERY MINOR ethics concerns only"]

**Final Justification:**

The authors have addressed my initial concerns. So, I maintain my original assessment of the work.

**Limitations:**

yes

**Quality:**

3

**Strengths And Weaknesses:**

The paper does a nice job framing the problem, situating prior work, pointing out its limitations, and explaining the contribution clearly. This feels like a meaningful theoretical step toward understanding a practical method. I’m not an expert in this area, but the proof of Theorem 4.1 definitely doesn’t look routine. The sketch was helpful too: after applying some triangle inequalities, the key step becomes controlling uniform convergence over 1-Lipschitz functions under Gaussian perturbations, which his is handled using Gaussian complexity, and Wasserstein distance between a measure and its empirical version. That seems nontrivial.

On the downside, the introduction says that PE is competitive with—or even outperforms—state-of-the-art models in terms of FID and downstream task performance. But the algorithm in this paper isn’t exactly that practical version—it’s a variant. So it would’ve been useful to compare Algorithm 2 with the real-world PE.

---

> ### Author Rebuttal · Authors · 2025-07-31
>
> Thanks for your review. Below we address your questions and concerns.
>
> >` I’m a little confused about the implication of Lemma 3.1. There is no qualifier on $\mathcal A}$. Does this mean the lower bound applies to any algorithm? If so, why doesn’t it apply to your algorithm? Or maybe you meant refers specifically to Algorithm 1?`
>
> The implication of Lemma 1 does apply for any algorithm ${\mathcal A}$ such that its output $\mathcal A(S)$ is $\eta$-close to $S$. Note that if $S$ and ${\mathcal A}(S)$ are $\eta$-close, then $W_1(\mathcal A(S),S) \leq \eta$. While $\eta$-closeness is sufficient to obtain a bound on the $W_1$ distance, it is not necessary. The reason our algorithm does not suffer from this lower bound is that the convergence of our variant of PE does not rely on $\eta$-closeness arguments.
>
> >`Also, just a minor formatting thing, but in Algorithm 1, the line is highlighted in blue, while the same step in Algorithm 2 isn’t. Was that intentional?`
>
> In algorithm 2 we have $S_0 \leftarrow$ RANDOM\_API(n_s) instead of $S_0 \leftarrow$RANDOM\_API(n). While subtle, this is a difference between our work and the previous theoretical work on PE: previously you were forced to create $n$ synthetic datapoints in order to use $\eta$-closeness arguments, while our framework allows to vary the number of synthetic samples (see figure 3 for the effects of picking $n_s$ suboptimally). We will highlight this point.
>
> >`the algorithm in this paper isn’t exactly that practical version—it’s a variant. So it would’ve been useful to compare Algorithm 2 with the real-world PE.`
>
> The differences between PE in practice and our variant are the $D_{BL}$ projection step and the API choice. Our variant is less practical but much more amenable to theoretical analysis. We will compare more explicitly both algorithms in the paper.

---

> > ### Comment · Reviewer_SoiT · 2025-08-01
> >
> > I thank the authors for clarification. I will keep my original score.

---

### Official Review · Reviewer_Kaym · 2025-06-28

**Clarity:** 4
**Significance:** 3
**Originality:** 3
**Rating:** 5
**Confidence:** 5

**Summary:**

### Summary

This paper provides novel worst-case theoretical analysis for the Private Evolution (PE) algorithm, a promising training-free method for differentially private (DP) synthetic data generation. PE iteratively refines an initial synthetic dataset (generated independently of private data) by creating variations and privately voting for those closest to the private dataset.

Previous theoretical analyses of PE relied on highly unrealistic assumptions (e.g., every private data point is duplicated many times), making those convergence guarantees inapplicable in practice. This paper carefully analyzes these limitations, proving lower bounds that show why such assumptions are necessary for prior proofs.

The authors then propose a new theoretical variant of PE that better resembles the practical implementation. The authors prove worst-case convergence guarantees in terms of expected 1-Wasserstein distance under $(\varepsilon, \delta)$-DP, showing that PE's error decays as roughly $O(d(n\varepsilon)^{-1/d})$. They also generalize parts of their analysis to Banach spaces and establish connections to the Private Signed Measure Mechanism (PSMM).

The paper includes simulations on synthetic datasets to validate and illustrate the theory, showing how PE’s convergence depends on initialization and parameter choices.

---

### Contributions

The main conceptual contribution is a more principled variant of PE that better resembles the practical implementation of PE and also lends itself well to worst-case theoretical analysis.
Some experiments on synthetic datasets verify the worst-case rates.

Some beyond worst-case analysis is also developed for "well-clustered" datasets, but it is unclear if the assumptions are realistic in practice.

A connection is also made to the private signed measure mechanism (PSMM) to interpret PE as an iterative version of PSMM.

---

### Techniques

The authors show that the variation step of PE contracts the $W_1$ distance to the dataset in expectation while carefully controlling the error introduced in the private voting step.
The analysis of the variation step seems to be mostly taken from [31].
The private voting step uses empirical process theory, which, as the authors mention, resembles [24].

---

#### References

[24] Y. He, R. Vershynin, and Y. Zhu. Algorithmically effective differentially private synthetic data. COLT'23

 [31] Z. Lin, S. Gopi, J. Kulkarni, H. Nori, and S. Yekhanin. Differentially private synthetic data via foundation model APIs 1: Images. ICLR'24

**Questions:**

### Main Questions
1. How reasonable is the current definition of VARIATION_API?
1. Is the worst-case dependence on dimension reflected in practice? It would be interesting to see if the dependence on dimension matches even on synthetic datasets.

---

### Minor Nitpicks
1. The current proof of Theorem 4.1 (main theorem) might need to restrict $\sigma$; the number of synthetic points $n_S\propto \sigma^{-1}$ wouldn't make sense for $\sigma$ arbitrarily large.
1. The domain $\Omega$ should probably be taken to be closed and convex, otherwise the projection of the VARIATION_API does not necessarily shrink the distance from the unprojected point (Lemma D.1 Case 2)
1. The statement of Lemma D.2 is intuitive, but its proof is notationally dense. A summary of the idea might help readers.
1. Is the term $C$ in the statement of Lemma D.4 an absolute constant? Please state so if it is.
1. Lemma D.4 is a nice application of chaining, but I wonder if a one-step discretization bound already suffices? Chaining usually helps to remove some log factors to achieve a sharp analysis but this doesn't seem to apply here.
1. The proof of Lemma D.4 (Line 939) doesn't seem to go through as stated. For an arbitrary pair $f_k\in T_k, f_{k-1}\in T_{k-1}$, their value can vary significantly. Indeed, the current proof uses the fact that there is a fixed function $f$ that is close to both. But this is not guaranteed for all pairs. Some slightly smarter choice of $f_k, f_{k-1}$'s should still work.
1. The $(5/\beta)^{d/2}$ term in the integral of the proof of Corollary D.1 (Line 951) should be $(5{\color{red}D}/\beta)^{d/2}$.
1. I believe Lemma D.5 can be proved via chaining and analyzing the Rademacher complexity of $\mathcal{F}_{BL}$. Since the Gaussian complexity already upperbounds this quantity, it might make sense to briefly mention this fact to make the paper even more self-contained.

**Ethical Concerns:**

["NO or VERY MINOR ethics concerns only"]

**Final Justification:**

The authors committed to providing a clearer framing of their theoretical contributions with respect to the practical implications. This addressed my main concern, and I maintain my score.

**Limitations:**

The authors claim that the only difference from practical PE is the projection in bounded Lipschitz distance. However, it is unclear if the current definition of VARIATION_API is a realistic model of the foundation model APIs applied in practice.

To be clear, I believe this work is a solid theoretical contribution.
However, it is presented as an attempt to bridge theory and practice and thus merits a discussion of this issue.

**Quality:**

3

**Strengths And Weaknesses:**

### Strengths

1. The main strength is a careful, rigorous analysis of PE’s convergence with more realistic assumptions.
This is important as PE is popular and seems to be quite practical.

1. The exposition is clear, and the proof overview in the main body is very intuitive. The proofs included in the appendix are also well-structured and include a nice application of chaining.

---

### Weaknesses

1. While the paper makes progress towards bridging theory and practice of PE, a major assumption that was not discussed at all is the theoretical formulation of the variation step, which is defined as taking random spherical Gaussian steps from the current synthetic data point, with variance chosen at different scales. While it may be very difficult to theoretically model the variation step of the actual PE implementation, as it relies on calling foundation model APIs, a paper that attempts to eliminate unrealistic assumptions should at least discuss this definition.

1. While the algorithmic variation is interesting, the analysis mainly consists of (cleverly) bringing together existing techniques.

1. The experiments do not vary the dimension and are performed in $\mathbb{R}^2$. This cannot confirm how accurate the theoretical predictions are for higher dimensional settings, which is the main regime of interest these days.

1. There are some small inconsistencies in the proofs within the appendix which should be easily addressable; I include the ones I found in the Questions section.

---

> ### Author Rebuttal · Authors · 2025-07-31
>
> Thank you for your detailed review and constructive comments. Below we address your concerns and questions.
>
> >`How reasonable is the current definition of VARIATION_API? It is unclear if the current definition of VARIATION_API is a realistic model of the foundation model APIs applied in practice.`
>
> We believe that our definition of VARIATION_API is reasonable. Adding noise at different scales can be seen as an attempt to model the different degrees of variation of the VARIATION_API. For example, the number of backward/forward steps of a diffusion model, or the amount of noise added at each step, affects the degree of similarity of the synthetic image to the original one. While one may challenge the Gaussianity assumption, the analogy to our modeling choice should at least be intuitively clear. We agree that future work could try to analyze more complex noise distributions that may be closer to the APIs used in practice, but we think our overall schema for modeling and analysis can likely still provide a blueprint for such a refined analysis. For amenable theoretical analysis, some analytically tractable approximations are required, which motivated our specific choice, which was also the choice made in the previous PE theoretical analysis in [1]. We will add this as a limitation of our work, and remark that while we took an important first step towards bridging theory and practice of PE, there is still room for improvement in the modeling of the APIs.
>
> >`Is the worst-case dependence on dimension reflected in practice? It would be interesting to see if the dependence on dimension matches even on synthetic datasets.`
>
> In synthetic experiments, we previously varied dimension $d$ from 1 to 10 and observed that error grows with the dimension when fixing the rest of the parameters. We will add these results to the Appendix.
>
> >`Minor Nitpicks`
>
> Thanks for catching these!
>
> (1) We will restrict $n_s$ to be the maximum between the current definition and 1, which should lead to $W_1$ error of $\min\{dD\sigma^{1/d}, D\}$ in the main theorem. (2) We consider a domain that is compact and convex. (3) We'll add an explanation of Lemma D.2. (4) We will say $C$ is an absolute constant in Lemma D.4. (5) We believe one-step discretization loses poly(n) factors as compared to chaining. (6) The supremum will be taken appropriately; the argument still goes through. (7) The term $D$ that was missing has been corrected. (8) We will carefully verify the reviewer's claim and definitively add it to the paper if correct.
>
> [1] Lin, Zinan, et al. "Differentially private synthetic data via foundation model apis 1: Images." arXiv preprint arXiv:2305.15560 (2023).

---

> > ### Comment · Reviewer_Kaym · 2025-08-02
> >
> > Thanks for the reply. I believe this is a nice contribution, once more discussion of the model/assumptions/limitations (that other reviewers also mentioned) is included. I maintain my score and look forward to reading the final version.

---

### Official Review · Reviewer_MgMH · 2025-07-01

**Clarity:** 4
**Significance:** 3
**Originality:** 3
**Rating:** 5
**Confidence:** 3

**Summary:**

Private evolution (PE) is a recently developed method for generating private synthetic data from generative AI systems. PE algorithms work by generating random initial data then iteratively re-sampling them to be more similar to a private dataset while respecting differential privacy. These methods have shown promise, particularly in the text and image domains but prior to this work there has only been one work that attempts to analyze PE and give a convergence result. The authors of this work show that the previous analysis of PE depends on several limiting assumptions and that the PE algorithm they analyze is significantly different from the PE algorithms that are used in practice. This paper analyzes a version of PE that is similar to practical PE algorithms and shows and gives both a worst case convergence guarantee and a data-dependent bound in terms of Wasserstein 1 distance. This new analysis also connects to the existing PSMM method and shows how it is related to PE.

**Questions:**

The authors model the variation api as being a noise addition step, it is unclear how accurate of an approximation this is for what generative models do when asked to create variations of an image. What kind of conditions are necessary for the variation api to meet to make this analysis work?

Is the noise scale for the variation api given in section 3 optimal under both the previous analysis and your analysis. Do you believe that practitioners should be tweaking their variation api to target a particular amount of variation?

**Ethical Concerns:**

["NO or VERY MINOR ethics concerns only"]

**Final Justification:**

My suggestions and questions were answered by the authors, and I believe that the changes in language that the authors agreed to will improve the manuscript. I feel that my original score of 5 was and still is appropriate based on the foreseeable impact of this work.

**Limitations:**

I believe the authors overstate how similar their theoretical PE is to practical PE, especially w.r.t. how they represent the variation api, this should be discussed as a limitation.

**Paper Formatting Concerns:**

I do not have any concerns about the formatting of the paper

**Quality:**

3

**Strengths And Weaknesses:**

Strengths:

The paper is written well and does a good job of giving the main proof ideas in the main text so that readers can understand the contribution at a high level before reading the full details in the supplementary material.

The authors clearly explain the assumptions made by the previous analysis of PE and give a convincing lower bound showing that those assumptions are necessary for their proof approach.

The proof techniques developed by the authors give reasonable upper bounds with respect to a version of PE that is more similar to the PE algorithms that are used in practice.

The connection between PE and the Private Signed Measure Mechanism (PSMM) is clear and gives a useful lens for understanding PE.

Weaknesses:

The authors claim that their analysis gives useful information for practitioners who are trying to utilize PE (how to select the number of rounds and number of synthetic data points). The support for this empirical claim is all with respect to simulation studies where the data are points in a ball within R2. This is very different from the domains where the mechanism is actually being used in practice (text and images) so it is unclear if the insights from their analysis will transfer to these domains.

---

> ### Author Rebuttal · Authors · 2025-07-31
>
> Thank you for your review. Below we address your concerns and questions.
>
> >`The authors claim that their analysis gives useful information for practitioners who are trying to utilize PE (how to select the number of rounds and number of synthetic data points). The support for this empirical claim is all with respect to simulation studies where the data are points in a ball within R2. This is very different from the domains where the mechanism is actually being used in practice (text and images) so it is unclear if the insights from their analysis will transfer to these domains.`
>
>
> We claimed that our theory can guide the practice of PE and illustrated this with experiments in Section 6. We were able to guide the practice of our variant of PE because we studied it carefully, but we will clarify that this does not imply that every time a variant of PE is run in practice the parameter setting should be used. We will soften the language about connections to practice: the APIs used in practical PE variants are much harder to analyze theoretically, and we leave as an interesting future direction to understand the utility of PE when these are used.
>
> >`The authors model the variation api as being a noise addition step, it is unclear how accurate of an approximation this is for what generative models do when asked to create variations of an image. What kind of conditions are necessary for the variation api to meet to make this analysis work?`
>
> Adding noise at different scales can be seen as an attempt to model the different degrees of variation of the VARIATION_API. For example, the number of backward/forward steps of a diffusion model, or the amount of noise added at each step, affects the degree of similarity of the synthetic image to the original one. While one may challenge the Gaussianity assumption, the analogy to our modeling choice should at least be intuitively clear. We agree that future work could try to analyze more complex noise distributions that may be closer to the APIs used in practice, but we think our overall schema for modeling and analysis can likely still provide a blueprint for such a refined analysis. For amenable theoretical analysis, some analytically tractable approximations are required, which motivated our specific choice, which was also the choice made in the previous (impractical) PE theoretical analysis in [1].
>
> We will be more explicit about these observations in the paper and will add as future work to study the convergence of PE with different theoretically tractable APIs; for example, if we asume the API is a diffusion model, we could consider the variations as random variables arising from the discretization of certain diffusion processes underlying the model.
>
> >`Is the noise scale for the variation api given in section 3 optimal under both the previous analysis and your analysis`
>
> Our choice of noise scale optimally trades off two opposing terms in the proof of Lemma D.4, and it leads to an optimal rate in the leading order terms of $n,d,\epsilon$, and in that sense is optimal. (And yes, it essentially matches the noise scale in the previous analysis, also obtained by trading off opposing terms.)
>
>
> >`Do you believe that practitioners should be tweaking their variation api to target a particular amount of variation?`
>
> Our algorithm and analysis suggest that we should not be adding noise only at a single scale but across multiple scales, and so in that sense the answer to your question would be no.
>
>
>
> >`I believe the authors overstate how similar their theoretical PE is to practical PE, especially w.r.t. how they represent the variation api, this should be discussed as a limitation.`
>
> Thank you for pointing this out; we will add a paragraph on limitations of our work emphasizing that our variant of PE is not exactly the same as the one run in practice, but rather the meta-algorithm is almost the same one. Our analysis lays a foundation for future work to expand the set of APIs under which provable convergence is possible. We will also clarify these points in the introduction and abstract.
>
> References:
>
> [1] Lin, Zinan, et al. "Differentially private synthetic data via foundation model apis 1: Images." arXiv preprint arXiv:2305.15560 (2023).

---

### Official Review · Reviewer_iicN · 2025-07-12

**Clarity:** 4
**Significance:** 3
**Originality:** 3
**Rating:** 5
**Confidence:** 2

**Summary:**

The authors provide new theoretical results supporting privacy evolution

**Questions:**

It would be good to discuss scalability issues to high-dimensional data, or for small / large data sets.

**Ethical Concerns:**

["NO or VERY MINOR ethics concerns only"]

**Limitations:**

yes

**Quality:**

3

**Strengths And Weaknesses:**

Theoretical support for existing approaches is a positive aspect. It was difficult to see what affect the results have in practical terms, especially for high dimensional data.

---

> ### Author Rebuttal · Authors · 2025-07-31
>
> Thanks for the review.
>
> >`It was difficult to see what affect the results have in practical terms, especially for high dimensional data. It would be good to discuss scalability issues to high-dimensional data, or for small / large data sets.`
>
> 1) We agree that our variant of PE is not directly practical in high dimensions. It has been proven that under DP constraints, any computationally efficient algorithm cannot accurately preserve all marginals [1] and $W_1$ not only preserves all marginals, it preserves all Lipschitz queries. There may be metrics that do not suffer from the curse of dimensionality, but for $W_1$ this price is inherent and not an issue specific to our proposed variant of PE.
>
>     However, this point is counterbalanced between the large theory-practice gap in the PE literature, where the practical algorithms are very far from the sole variant for which theory exists. In this light, we narrow the gap, and we  theoretically analyze a variant of PE that is much closer to what people run in practice.
>
> 2) With regards to scalability, we believe that the theoretical insights gained from our paper could still be useful to understand other variants of PE that pose more structure. For example, the if the intrinsic dimension of the data in practice is much smaller than the ambient dimension, then our rates should only pay for the intrinsic dimension.
>
> ### References:
>
> [1] Ullman, Jonathan, and Salil Vadhan. "PCPs and the hardness of generating private synthetic data." Theory of Cryptography Conference.

---

### Decision · Program_Chairs · 2025-09-17

**Decision:**

Accept (poster)

**Comment:**

Private evolution (PE) is a recently proposed method that generates private synthetic data from generative AI systems by iteratively refining an initial synthetic dataset by creating variations and privately voting for those closest to the private dataset. The authors of this work show that the previous analysis of PE depends on several limiting assumptions, which are often not reflected in practice. This paper analyzes a version of PE that is similar to practical PE algorithms and shows a worst case convergence guarantee for the Wasserstein distance given the number of samples.

Due to the high variance in the reviewer feedback, this paper received substantial deliberation in both the Reviewer-AC and the AC-SAC discussion phases. Although there were concerns about the practical performance of private evolution in specific domains, reviewers ultimately agreed that theoretical groundwork is valuable, even before the technology is fully mature.

Additionally, reviewers expressed concerns that this paper proves convergence in terms of the number of samples $n$, whereas the previous paper proves convergence in terms of the number of iterations $T$ (though there is also a somewhat implicit dependency on $n$). Thus, there is a mismatch between the convergence guarantees, which compromises the paper's main goal of filling gaps in previous literature. On the other hand, if the convergence guarantees are viewed strictly in terms of sample complexity and learning theory, then the results are meaningful in light of existing lower bounds for Wasserstein distance that are exponential in $d$.

Action item: Clarify the context of the convergence guarantees with respect to sample complexity and the corresponding implications for the number of iterations, if relevant.